# SKETCHED ADAPTIVE FEDERATED DEEP LEARNING: A SHARP CONVERGENCE ANALYSIS

## ABSTRACT

Combining gradient sketching methods (e.g., CountSketch (Charikar et al., 2002; Rothchild et al., 2020), quantization (Tang et al., 2021)) and adaptive optimizers (e.g., Adam (Kingma & Ba, 2014), AMSGrad (Reddi et al., 2019)) is a desirable goal in federated learning (FL), with potential benefits on both fewer communication rounds and smaller per-round communication. In spite of the preliminary empirical success of sketched adaptive methods, existing convergence analyses show the communication cost to have a linear dependence on the ambient dimension (Spring et al., 2019; Tang et al., 2021), i.e., number of parameters, which is prohibitively high for modern deep learning models.

In this work, we introduce specific sketched adaptive federated learning (SAFL) algorithms and, as our main contribution, provide theoretical convergence analyses in different FL settings with guarantees on communication cost depending only logarithmically (instead of linearly) on the ambient dimension. Unlike existing analyses, we show that the entry-wise sketching noise existent in the preconditioners and the first moments of SAFL can be implicitly addressed by leveraging the recently-popularized anisotropic curvatures in deep learning losses, e.g., fast decaying loss Hessian eigen-values. In the i.i.d. client setting of FL, we show that SAFL achieves $O(1/\sqrt{T})$ convergence, and $O(1/T)$ convergence near initialization. In the non-i.i.d. client setting, where non-adaptive methods lack convergence guarantees, we show that SACFL (SAFL with clipping) algorithms can provably converge in spite of the additional heavy-tailed noise. Our theoretical claims are supported by empirical studies on vision and language tasks, and in both fine-tuning and training-from-scratch regimes. Surprisingly, as a by-product of our analysis, the proposed SAFL methods are competitive with the state-of-the-art communication-efficient federated learning algorithms based on error feedback.

## 1 INTRODUCTION

Despite the recent success of federated learning (FL), the cost of communication arguably remains the main challenge. (Wang et al., 2023) showed that a 20 Gbps network bandwidth is necessary to bring the communication overhead to a suitable scale for finetuning GPT-J-6B, which is unrealistic in distributed settings. Even with good network conditions, reduction on the communication complexity means one can train much larger models given the same communication budget.

The communication cost of FL can be represented as $O(dT)$, where $d$ is the ambient dimension of the parameter space and $T$ is the number of rounds for convergence. Various methods have been proposed to minimize $T$, e.g., local training (Stich, 2018), large batch training (Xu et al., 2023). Folklores in centralized training regimes suggest that $T$ heavily relies on the choice of optimizers, where adaptive methods usually demonstrate faster convergence and better generalization performance, especially in transformer-based machine learning models (Reddi et al., 2019). In decentralized settings, adaptive methods are also favorable due to their robustness to data heterogeneity, e.g., adaptive methods are guaranteed to converge under heavy-tailed noise while SGD does not (Zhang et al., 2020). These favorable merits, in principle, should be preserved in communication-efficient FL algorithms.

The alternative approach of reducing communication costs is to be more thrifty on the communication bits at a single round, i.e., to reduce the $O(d)$ factor, which is dominant in the communication complexity for modern neural networks where $d \gg T$. Considerable efforts have been devoted

to design efficient gradient compression methods. Popular gradient compression methods include quantization (Alistarh et al., 2017; Chen et al., 2023; Reisizadeh et al., 2020; Liu et al., 2023a), sparsification (Alistarh et al., 2018; Wu et al., 2018; Rothchild et al., 2020) and sketching (Spring et al., 2019; Jiang et al., 2024; Song et al., 2023). However, most of these developments do not use adaptive methods, which involve anisotropic and nonlinear updates, and there are no easy ways to do error feedback in case of discrepancies. Indeed, the design and analysis of communication-efficient adaptive FL algorithms poses non-trivial challenges.

In this work, we first introduce a family of Sketched Adaptive FL (SAFL) algorithms, with flexibility on the choice of sketching methods and adaptive optimizers, that simultaneously accelerates convergence and reduces per round bits towards improved communication efficiency. At a high level, SAFL algorithms are in principle analogous to previous attempts (Tang et al., 2021; Chen et al., 2022; Wang et al., 2022), which showed preliminary empirical success of applying gradient compression with adaptive optimizers in homogeneous data scenarios. [ab: with those two previous papers?]Our SAFL algorithm adopts unbiased gradient estimators and hence eliminates the needs for error feedbacks. The choice of gradient estimators, which is a linear operator, in SAFL also avoids the extra round of compression on the server side required by sparsification (Stich et al., 2018) and quantization (Reisizadeh et al., 2020).

Despite the preliminary empirical success of combining gradient sketching and adaptive optimizers for federated deep learning, theoretical understanding on the promise of such algorithms is limited. Existing works on the theory has arguably alarming results, which do not match practice. For example, some results show that the iterations $T$ needed for convergence can be inversely proportional to the compression rate (Chen et al., 2022; Song et al., 2023). For constant per-round communication bits, the bounds indicate the iteration complexity to scale as $O(d)$, i.e., linearly with the ambient dimensionality, which is prohibitively large for modern deep learning models. The mismatch between potential issues in theory vs. preliminary empirical promise and possibly also not having precise forms of such algorithms for different FL scenarios may be preventing wide adoption of such algorithms.

As a major contribution of our current work, we provide theoretical guarantees of the proposed SAFL algorithms on the convergence rate in common FL scenarios (almost i.i.d. as well as heavy tailed), which depends only on a logarithmically (instead of linearly) on the ambient dimension $d$. The central technical challenge in addressing the dimensional dependence is to handle the entry-wise sketching noise in both the preconditioners and the first moments of the adaptive optimizers, which has been acknowledged non-trivial (Tang et al., 2021; Wang et al., 2022). Our sharper analysis leverages recent observations regarding the eigenspectrum structure of the loss Hessian in deep learning, which show the eigenvalues to be sharply decaying, with most eigenvalues being close to zero (Ghorbani et al., 2019; Zhang et al., 2020; Li et al., 2020; Liao & Mahoney, 2021; Liu et al., 2023b), and even conforming with a power-law decay (Xie et al., 2022; Zhang et al., 2024), while the conventional smoothness conditions assume uniform curvature in all directions which can be overly pessimistic in the context of deep learning. This specific eigenspectrum structure provides significant advantages in the sharp analysis of sketching noise in adaptive methods. Our work leverages such geometric structure, leading to the following main contributions:

(1) For the benign almost i.i.d. FL setting, we introduce the sketched adaptive FL (SAFL) framework which incorporates random sketching techniques into adaptive methods. While the preconditoner in adaptive methods morphs the shape of sketching noise, which poses challenges in leveraging the anisotropic Hessian structure, we prove that the proposed sketching method effectively balances iteration complexity and sketching dimension $b$. We derive a high probability bound showing that a sketch size of $b = O(\log d)$ suffices to achieve an asymptotic $O(1/\sqrt{T})$ dimension-independent convergence rate in non-convex deep learning settings.

(2) For the heavy-tailed noise common in data-heterogeneous FL, where non-adaptive methods are not guaranteed to converge, we propose the Sketched Adaptive Clipped Federated Learning (SACFL) which guarantees the boundedness of the second moments. We theoretically show that SACFL can achieve optimal convergence rate under $\alpha$-moment noise with $\alpha \in (1, 2]$, regardless of the extra noise introduced by random sketching.

(3) We validate our theoretical claims with empirical evidence on deep learning models from vision (ResNet, Vision Transformer) and language (BERT) tasks. We cover both fine-tuning and training-from-scratch regimes. The proposed SAFL algorithm achieves comparable performance with

---

**Algorithm 1** Sketched Adaptive Federated Learning ( SAFL , SACFL )

---

**Input:** Learning rate $\eta$, initial parameters $x_0$, optimizer ADA_OPT( AMSGrad, Adam , AdaClip )

**Output:** Updated parameters $x_T$

Initialize server moments: $m_0 = 0, v_0 = 0, \hat{v}_0 = 0$, client moments: $m_0^c = 0, v_0^c = 0, \hat{v}_0^c = 0$,

client initial parameters: $x_{0,0}^c = x_0, \forall c \in [C]$;

**for** $t = 1, 2, \ldots, T$ **do**

    **Client Updates:**

    **for** $c = 1, 2, \ldots, C$ **do**

        Client model synchronization:
$$x_{t,0}^c, m_t^c, v_t^c, \hat{v}_t^c = \text{ADA\_OPT}(x_{t-1,0}^c, m_{t-1}^c, v_{t-1}^c, \hat{v}_{t-1}^c, \bar{m}_t, \bar{v}_t);$$

        **for** $k = 1, 2, \ldots, K$ **do**

            Compute stochastic gradient $g_{t,k-1}^c$ with respect to the parameters $x_{t,k-1}^c$;

            Perform a single gradient step: $x_{t,k}^c = x_{t,k-1}^c - \eta_t g_{t,k-1}^c$;

        **end**

        Sketch (compress) the parameter updates:
$$\bar{m}_t^c = \text{sk}(x_{t,0}^c - x_{t,K}^c); \qquad \bar{v}_t^c = \|x_{t,K}^c - x_{t,0}^c\|;$$

    **end**

    **Server Updates:**

    Elementwise square as second moments: $\bar{v}_t^c = (\bar{m}_t^c)^2, \ \forall c \in [C]$;

    Average sketched client updates, second moments and send back to clients
$$\bar{m}_t = \frac{1}{C} \sum_{c=1}^{C} \bar{m}_t^c; \qquad \bar{v}_t = \frac{1}{C} \sum_{c=1}^{C} \bar{v}_t^c;$$

    Update paramters and moments: $x_t, m_t, v_t, \hat{v}_t = \text{ADA\_OPT}(x_{t-1}, m_{t-1}, v_{t-1}, \hat{v}_{t-1}, \bar{m}_t, \bar{v}_t)$.

**end**

---

the full-dimensional unsketched adaptive optimizers, and are competitive with the state-of-the-art communication-efficient federated learning algorithms based on error feedback. We also validate the SACFL algorithms can achieve similar performance as the unsketched clipping algorithm when the local client gradient norms are $\alpha$-stable heavy-tailed. [ab: what is the take-away for the heterogenous case? performs same as unsketched?]

## 2 SKETCHED ADAPTIVE FL UNDER MILD NOISE

In this section, we consider federated learning on nearly-i.i.d client data distribution. The objective is to develop communication-efficient adaptive learning algorithms. We will first propose the general framework for applying gradient compression to FedOPT (Reddi et al., 2020), and proceed with the mild-noise assumptions and convergence analysis of the algorithm.

### 2.1 SKETCHED ADAPTIVE FL (SAFL)

A canonical federated learning setting involves $C$ clients, each associated with a local data distribution $\mathcal{D}_c$. The goal is to minimize the averaged empirical risks: $\mathcal{L}(x) = \frac{1}{C} \sum_{c=1}^{C} \mathbb{E}_{\xi \sim \mathcal{D}_c} l(x, \xi)$, where $l$ is the loss function, $x \in \mathbb{R}^d$ is the parameter vector, and $\xi$ is the data sample. We denote $\mathcal{L}^c(x) = \mathbb{E}_{\xi \sim \mathcal{D}_c} l(x, \xi), c \in [C]$ as the client loss function computed over the local data distribution. We denote $g_{t,k}^c$ as the mini-batch gradient over $\mathcal{L}^c(x)$ at global step $t$ and local step $k$.

Algorithm 1 presents a generic framework of communication-efficient adaptive methods, which calls adaptive optimizers as subroutines. We focus on SAFL (calling Algorithm 2) in this section, and will move to SACFL (calling Algorithm 3) in Section 3. The two algorithms are highlighted for their unique procedures separately. SAFL ignores the highlighted sections of SACFL, and vice versa. In case of ambiguity, we also provide separate versions of the two algorithms in the appendix.

---

**Algorithm 2** ADA_OPT (AMSGrad)

---

**Input:** iterate $x_{t-1}$, moments $m_{t-1}, v_{t-1}, \hat{v}_{t-1}$, sketched updates $\bar{m}_t, \bar{v}_t$
**Parameter:** Learning rate $\kappa$, $\beta_1$, $\beta_2$, Small constant $\epsilon$
**Output:** Updated parameters $x_t$, and moments $m_t, v_t, \hat{v}_t$
Update first moment estimate: $m_t = \beta_1 \cdot m_{t-1} + (1 - \beta_1) \cdot \texttt{desk}(\bar{m}_t)$;
Update second moment estimate: $v_t = \beta_2 \cdot v_{t-1} + (1 - \beta_2) \cdot \texttt{desk}(\bar{v}_t)$;
Update maximum of past second moment estimates: $\hat{v}_t = \max(\hat{v}_{t-1}, v_t)$.
Update parameters: $x_{t+1} = x_t - \frac{\kappa}{\sqrt{\hat{v}_t} + \epsilon} \cdot m_t := x_t - \kappa \hat{V}_t^{-1/2} m_t$.

---

We denote $T$ as the total training rounds. At each round, after $K$ local training steps, client $c$ sends to the server the sketched local model updates with a sketching operator $\texttt{sk}: \mathbb{R}^d \to \mathbb{R}^b$. If $b \ll d$ without deteriorating the performance too much, the communication cost per round can be reduced from $O(d)$ to $O(b)$. The server retrieves lossy replicates of the updates and the second moments using a desketching operator $\texttt{desk}: \mathbb{R}^b \to \mathbb{R}^d$. The gradient compression steps differentiate Algorithm 1 from the subspace training methods (Gressmann et al., 2020; Wortsman et al., 2021) since we are utilizing the global gradient vector in each round rather than solely optimizing over the manifold predefined by a limited pool of parameters. The choice of server-side optimizers determines how the lossy replicates in $\mathbb{R}^d$ are used to update the running moments (i.e. the momentum and the second moments). The server sends the moments in $\mathbb{R}^b$ back to the clients so that each client can perform an identical update on its local model, which ensures synchronization as each training round starts.

**Remark 2.1.** *(Sketching Randomness).* At each single round, the sketching operators $\texttt{sk}$'s are shared among clients, via the same random seed, which is essential for projecting the local model updates to a shared low dimensional subspace and making direct averaging reasonable. On the other hand, we use different $\texttt{sk}$'s at different rounds so that the model updates lie in distinct subspaces.[ab: i did not understand the last part.] $\qquad\square$

## 2.2 RANDOM SKETCHING

We will first introduce the desired characteristics of compression and then list a family of sketching algorithms which possess those properties.

**Property 1.** *(Linearity). The compression operators are linear w.r.t the input vectors, i.e.* $\texttt{sk}(\sum_{i=1}^n v_i) = \sum_{i=1}^n \texttt{sk}(v_i)$ *and* $\texttt{desk}(\sum_{i=1}^n \bar{v}_i) = \sum_{i=1}^n \texttt{desk}(\bar{v}_i)$, $\forall \{v_i, \bar{v}_i \in \mathbb{R}^d\}_{i=1}^n$.

**Property 2.** *(Unbiased Estimation). For any vector $v \in \mathbb{R}^d$, $\mathbb{E}[\texttt{desk}(\texttt{sk}(v))] = v$.*

**Property 3.** *(Bounded Vector Products). For any fixed vector $v, h \in \mathbb{R}^d$, $\mathbb{P}(|\langle \texttt{desk}(\texttt{sk}(v)), h\rangle - \langle v, h\rangle| \geq (\frac{\log^{1.5}(d/\delta)}{\sqrt{b}})\|v\|\|h\|) \leq \Theta(\delta)$.*

Property 1 and 2 guarantee the average of first moments in Algorithm 1 over clients are, in expectation, the same as those in FedOPT. Property 3 quantifies the bound on the deviation of vector products when applying compression. $\texttt{sk}(v) = Rv$ and $\texttt{desk}(\bar{v}) = R^\top \bar{v}$, where $R \in \mathbb{R}^{b \times d}$ is a random sketching operator, satisfy all the properties above (Song et al., 2023). We denote $R_t$ as the sketching operator used in round $t$. Different instantiations of $R$ constitute a rich family of sketching operators, including i.i.d. isotropic Gaussian, Subsampled Randomized Hadamard Transform (SRHT) (Lu et al., 2013), and Count-Sketch (Charikar et al., 2002), among others. The specific error bounds for these special cases can be found in Appendices B.1, B.2, and B.3 respectively.

## 2.3 CONVERGENCE ANALYSIS

We first state a set of standard assumptions commonly used in the literature of first-order stochastic methods. We focus on the mild noise assumptions, which are typically observed when the training data are nearly i.i.d. over clients. We will use $\|\cdot\|$ to denote $L_2$-norm throughout the work.

**Assumption 1.** *(Bounded Global Gradients). Square norm of the gradient is uniformly bounded, i.e.,* $\|\nabla\mathcal{L}(x)\|^2 \leq G_g^2$.

**Assumption 2.** *(Bounded Client Gradients). For every client, there exists a constant $G_c \geq 0$, such that $\|\nabla\mathcal{L}^c(x)\|^2 \leq G_c^2$, $c \in [C]$.*

For simplicity, we define $G := \max\{\max\{G_c\}_{c=1}^{C}, G_g\}$ to denote the upper bound for client and global gradient norms. We also assume the stochastic noise from minibatches is sub-Gaussian, which is widely adopted in first-order optimization (Harvey et al., 2019; Mou et al., 2020).

**Assumption 3.** *(Sub-Gaussian Noise). The stochastic noise $\|\nabla\mathcal{L}^c(x) - g^c(x)\|$ at each client is a $\sigma$-sub-Gaussian random variable, i.e. $\mathbb{P}(\|\nabla\mathcal{L}^c(x) - g^c(x)\| \geq t) \leq 2\exp(-t^2/\sigma^2)$, for all $t \geq 0$.*

Besides, we have assumptions on the Hessian eigenspectrum $\{\lambda_i, v_i\}_{i=1}^{d}$ of the loss function $\mathcal{L}$.

**Assumption 4.** *(Hessian Matrix Eigenspectrum) The smoothness of the loss function $\mathcal{L}$, i.e. the largest eigenvalue of the loss Hessian $H_{\mathcal{L}}$ is bounded by L, $\max_i \lambda_i \leq L$. The sum of absolute values of $H_{\mathcal{L}}$ eigenspectrum is bounded by $\hat{L}$, i.e. $\sum_{i=1}^{d}|\lambda_i| \leq \hat{L}$. [ab: update to $\hat{L}$?]*

The assumption of bounded sum of eigenspectrum has been validated by several recent literatures, in the context of deep learning, where the eigenspectrum is observed to sharply decay (Ghorbani et al., 2019; Li et al., 2020; Liu et al., 2023b), have bulk parts concentrate at zero (Sagun et al., 2016; Liao & Mahoney, 2021) or conform with a power-law distribution (Xie et al., 2022; Zhang et al., 2024). We quote their plots in Appendix E for completeness. Our empirical verification under the setting of FL can also be found in Fig. 6 in Appendix E.

**Remark 2.2.** *(Three types of noises in Algorithm 1).* One of the key technical contributions of this work is to theoretically balance the noises of different sources and derive a reasonable convergence rate which is independent of the ambient dimension. The noise in the training process stems from the mini-batch training, the client data distribution, and the compression error due to sketching. The stochastic error of mini-batch training is sub-Gaussian by Assumption 3. We adopt $\delta_g$ to control the scale of the sub-Gaussian noise[ab: Assumption 3 uses $\sigma$, how is $\sigma$ related to $\delta_g$?]. The i.i.d. data distribution leads to the bounded gradient assumption (Assumption 2). The sketching error depends on the specific choice of sketching methods, but is always controlled by the bounded property on vector products (Property 3) with a universal notation $\delta$. All three kinds of noises are unbiased and additive to the gradient, though may have sequential dependencies. Therefore, for the analysis (Appendix C), we will introduce a martingale defined over the aggregated noise, using which we can derive a high-probability concentration bound for the variance. We denote $\nu$ as the tunable scale [ab: the $\psi_2$-norm of the subG martingale?][lu: let's discuss during meeting.]for the $\psi_2$-norm (Vershynin, 2018) in the martingale. □

Now we can derive the convergence analysis of Algorithm 1 as in Theorem 2.1. All technical proofs for this section are in Appendix C and we provide an outline of the proof techniques in Section 2.4.

**Theorem 2.1.** *Suppose the sequence of iterates $\{x_t\}_{t=1}^{T}$ is generated by Algorithm 1 (SAFL) with a constant learning rate $\eta_t \equiv \eta$. Under Assumptions 1-4, for any $T$ and $\epsilon > 0$, with probability $1 - \Theta(\delta) - O(\exp(-\Omega(\nu^2))) - \delta_g$,*

$$\left(\sqrt{1 + \frac{\log^{1.5}(CKd^2T^2/\delta)}{\sqrt{b}}}\eta KG + \epsilon\right)^{-1} \kappa\eta K \sum_{t=1}^{T}\|\nabla\mathcal{L}(x_t)\|^2 \leq \mathcal{L}(z_1) + \frac{1}{\epsilon}\kappa\eta^2 LK^2G^2T$$

$$+ \nu\kappa\eta K\sqrt{T}(\frac{\log^{1.5}(CKTd/\delta)}{\sqrt{b}}\frac{G^2}{\epsilon} + \frac{\sigma}{\epsilon}\log^{\frac{1}{2}}(\frac{2T}{\delta_g})) + \eta^2\kappa^2T(1 + \frac{\log^{1.5}(CKdT^2/\delta)}{\sqrt{b}})^2\frac{8}{(1-\beta_1)^2}\frac{\hat{L}K^2G^2}{\epsilon^2},$$

*where $\delta, \delta_g$, and $\nu$ are the randomness of sketching, sub-Gaussian noise, and martingales respectively.*

A non-asymptotic convergence bound of training with practical decaying learning rates can be found in Theorem C.2 in appendix. Given that we only introduce logarithmic factors on $d$ in the iteration complexity and the per-round communication $b$ is a constant, the total communication bits of training a deep model till convergence is also logarithmic w.r.t $d$. To better understand Theorem 2.1, we can investigate different regimes based on the training stages. For the asymptotic regime, where $T$ is sufficiently large, we can achieve an $O(1/\sqrt{T})$ convergence rate in Corollary 1.

**Corollary 1.** *With the same condition as in Thereom 2.1, for sufficiently large $T \geq \frac{G^2}{\epsilon^2}$, with probability $1 - \Theta(\delta) - O(\exp(-\Omega(\nu^2))) - \delta_g$,*

$$\frac{1}{T}\sum_{t=1}^{T}\|\nabla\mathcal{L}(x_t)\|^2 \leq \frac{2\mathcal{L}(z_1)\epsilon}{\kappa\sqrt{T}} + \frac{2}{\epsilon}\frac{LG^2}{\sqrt{T}} + \nu\frac{2}{\sqrt{T}}(\frac{\log^{1.5}(CKTd/\delta)}{\sqrt{b}}G^2 + \sigma\log^{\frac{1}{2}}(\frac{2T}{\delta_g}))$$

$$+ \kappa\frac{1}{\sqrt{T}}(1 + \frac{\log^{1.5}(CKdT^2/\delta)}{\sqrt{b}})^2\frac{16}{(1-\beta_1)^2}\frac{\hat{L}G^2}{\epsilon},$$

*where $\delta, \delta_g$ and $\nu$, are the randomness of sketching, sub-Gaussian noise and martingales respectively.*

[ab: add text to connect the two – AB will do] More interestingly, for the near-initialization regime, where $T$ is relatively small, we can observe that the coefficient of $\|\nabla\mathcal{L}(x_t)\|^2$ on the left hand side in Theorem 2.1 and C.2 is approximately a constant, given that $\epsilon$ is tiny. Therefore, SAFL can achieve an $O(1/T)$ convergence near initialization, which accounts for the empirical advantages over non-adaptive methods.

**Corollary 2.** *With the same condition as in Thereom 2.1, set $b \geq \log^3(CKd^2T^2/\delta)$ and constant $J_1 > \sqrt{2}G$, then for any $T \leq \frac{J_1 - \sqrt{2}G}{\epsilon^2}$, with probability $1 - \Theta(\delta) - O(\exp(-\Omega(\nu^2))) - \delta_g$,*

$$\frac{1}{J_1 T}\sum_{t=1}^{T}\|\nabla\mathcal{L}(x_t)\|^2 \leq \frac{\mathcal{L}(z_1)\epsilon}{\kappa T} + \frac{1}{\epsilon}\frac{LG^2}{T} + \frac{\nu}{T}(G^2 + \sigma\log^{\frac{1}{2}}(\frac{2T}{\delta_g})) + \frac{\kappa}{T}\frac{32}{(1-\beta_1)^2}\frac{\hat{L}G^2}{\epsilon},$$

*where $\delta, \delta_g$ and $\nu$, are the randomness of sketching, sub-Gaussian noise and martingales respectively.*

### 2.4 TECHNICAL RESULTS AND PROOF SKETCH

In this section, we provide a sketch of the proof techniques behind the main results. We focus on the proof of Theorem 2.1, and the proof of Theorem C.2 shares the main structure. The proof of Theorem 2.1 contains several critical components, which are unique to adaptive methods. We follow the common proof framework of adaptive optimization, and carefully deal with the noise introduced by random sketching in the momentum. We adopt AMSGrad (Alg. 2) as the server optimizer and it would be straightforward to extend the analysis to other adaptive methods.

We first introduce the descent lemma for AMSGrad. For conciseness, we denote the precondtioner matrix $\text{diag}((\sqrt{\hat{v}_t} + \epsilon)^2)$ as $\hat{V}_t$. Define an auxiliary variable $z_t = x_t + \frac{\beta_1}{1-\beta_1}(x_t - x_{t-1})$. The trajectory of $\mathcal{L}$ over $\{z_t\}_{t=1}^{T}$ can be tracked by the following lemma.

**Lemma 2.2.** *For any round $t \in [T]$, there exists function $\Phi_t \geq 0$ ,and $\Phi_0 \leq G$ such that*

$$\mathcal{L}(z_{t+1}) \leq \mathcal{L}(z_t) + \Phi_t - \Phi_{t+1} - \frac{\kappa\eta}{C}\sum_{c=1}^{C}\sum_{k=1}^{K}\nabla\mathcal{L}(x_t)^{\top}\hat{V}_{t-1}^{-1/2}R_t^{\top}R_t g_{t,k}^c + (z_t - x_t)^{\top}H_{\mathcal{L}}(\hat{z}_t)(z_{t+1} - z_t),$$

*where $H_{\mathcal{L}}(\hat{z}_t)$ is the loss Hessian at some $\hat{z}_t$ within the element-wise interval of $[x_t, z_t]$.*

Our objective henceforth is to bound the first-order descent term and the second-order quadratic term on the right hand side respectively.

**Second-Order Quadratic Term.** Denote $\{\lambda_j, v_j\}_{j=1}^{d}$ as the eigen-pairs of $H_{\mathcal{L}}(\hat{z}_t)$. The quadratic term can be written as $(z_t - x_t)^{\top}H_{\mathcal{L}}(\hat{z}_t)(z_{t+1} - z_t) = \sum_{j=1}^{d}\lambda_j\langle z_{t+1} - z_t, v_j\rangle\langle z_t - x_t, v_j\rangle$. The inner product terms can be viewed as a projection of the updates onto anisotropic bases. Since $z_{t+1} - z_t$ and $z_t - x_t$ can both be expressed by $x_{t+1} - x_t$ and $x_t - x_{t-1}$, we can bound the quadratic term using the following lemma.

**Lemma 2.3.** *For any $t \in [T]$, $|\langle x_t - x_{t-1}, v_j\rangle| \leq \kappa\eta(1 + \frac{\log^{1.5}(CKtd/\delta)}{\sqrt{b}})\frac{KG}{\epsilon}$, with probability $1 - \delta$.*

Bounding the inner-product term is non-trivial since $z_t$ contains momentum information which depends on the randomness of previous iterations. A proof of a generalized version of this statement is deferred to the appendix, where induction methods are used to address the dependence. Combining Lemma 2.3 with Assumption 4 yields a dimension-free bound on the second-order quadratic term.

**Remark 2.3.** A straightforward application of smoothness to the second-order term yields a quadratic term $\|R^{\top}Rg\|^2$, which is linearly proportional to $d$ in scale (Rothchild et al., 2020; Song et al., 2023). We avoid this dimension dependence by combining Property 3 of sketching and Assumption 4. $\square$

**First-Order Descent Term**. The first-order term in the descent lemma can be decomposed into three components, which we will handle separately:

$$\nabla\mathcal{L}(x_t)^\top \hat{V}_{t-1}^{-1/2} R_t^\top R_t g_{t,k}^c = \underbrace{\nabla\mathcal{L}(x_t)^\top \hat{V}_{t-1}^{-1/2}\nabla\mathcal{L}^c(x_t)}_{\mathcal{D}_1} + \underbrace{\nabla\mathcal{L}(x_t)^\top \hat{V}_{t-1}^{-1/2}(R_t^\top R_t g_{t,k}^c - \nabla\mathcal{L}^c(x_{t,k}^c))}_{\mathcal{D}_2}$$

$$+ \underbrace{\nabla\mathcal{L}(x_t)^\top \hat{V}_{t-1}^{-1/2}(\nabla\mathcal{L}^c(x_{t,k}^c) - \nabla\mathcal{L}^c(x_t))}_{\mathcal{D}_3}.$$

First, $\mathcal{D}_3$ can be reduced to a second-order term by smoothness over $\mathcal{L}$, $\nabla\mathcal{L}(x_t)^\top \hat{V}_{t-1}^{-1/2}(\nabla\mathcal{L}^c(x_{t,k}^c) - \nabla\mathcal{L}^c(x_t)) = -\eta\sum_{\tau=1}^{k}\nabla\mathcal{L}(x_t)^\top \hat{V}_{t-1}^{-1/2}\hat{H}_\mathcal{L}^c g_{t,\tau}^c$. Note that this term does not involve any stochasticity from random sketching, hence we can directly derive the upper bound by Cauchy-Schwartz. Next, since $\frac{1}{C}\sum_{c=1}^{C}\nabla\mathcal{L}^c(x_t) = \nabla\mathcal{L}(x_t)$, $\mathcal{D}_1$ composes a scaled squared gradient norm. Applying element-wise high probability bound on random sketching yields the lower bound for the scale.

**Lemma 2.4.** *For $\hat{V}_{t-1}^{-1/2}$ generated by Algorithm 1 (SAFL), with probability $1-\delta$,*

$$\nabla\mathcal{L}(x_t)^\top \hat{V}_{t-1}^{-1/2}\nabla\mathcal{L}(x_t) \geq \left(\sqrt{1 + \frac{\log^{1.5}(CKtd^2/\delta)}{\sqrt{b}}}\eta KG + \epsilon\right)^{-1}\|\nabla\mathcal{L}(x_t)\|^2.$$

**Martingale for zero-centered noise.** $\mathcal{D}_2$ contains a zero-centered noise term $R_t^\top R_t g_{t,k}^c - \nabla\mathcal{L}^c(x_{t,k}^c)$, where the randomness is over $R_t$ and the mini-batch noise at round $t$. Although $x_{t,k}^c$ has temporal dependence, the fresh noise due to mini-batching and sketching-desketching at round $t$ is independent of the randomness in the previous iterations. Therefore, the random process defined by the aggregation of the zero-centered noise terms over time forms a martingale. The martingale difference can be bounded with high probability under our proposed sketching method. Then by adapting Azuma's inequality on a sub-Gaussian martingale, we have

**Lemma 2.5.** *With probability $1 - O(\exp(-\Omega(\nu^2))) - \delta - \delta_g$,*

$$\sum_{t=1}^{T}\left|\frac{1}{C}\sum_{c=1}^{C}\sum_{k=1}^{K}\nabla\mathcal{L}(x_t)^\top \hat{V}_{t-1}^{-1/2}(R_t^\top R_t g_{t,k}^c - \nabla\mathcal{L}^c(x_{t,k}^c))\right| \leq \nu\sqrt{T}(\frac{\log^{1.5}(CKTd/\delta)}{\sqrt{b}}\frac{KG^2}{\epsilon} + \frac{\sigma}{\epsilon}\log^{\frac{1}{2}}(\frac{2T}{\delta_g})).$$

Finally, applying union bounds to these parts and telescoping the descent lemma leads to Theorem 2.1.

## 3 SKETCHED ADAPTIVE CLIPPED FL FOR HEAVY-TAILED NOISE

In this section, we study the performance of Sketched Adaptive Clipped FL (SACFL) defined in Algorithm 1 calling Algorithm 3 in the context of heavy-tailed noise over gradient norms. This is arguably the more challenging setting and requires carefully addressing the noises with clipping.

### 3.1 HEAVY-TAILED NOISE AND SKETCHED CLIPPING-BASED ADAPTIVE METHODS

We start with the key bounded $\alpha$-moment assumption for the heavy-tailed stochastic first-order oracle.

**Assumption 5.** *(Bounded $\alpha$-Moment). There exists a real number $\alpha \in (1, 2]$ and a constant $G \geq 0$, such that $\mathbb{E}[\|\nabla\mathcal{L}^c(x, \xi)\|^\alpha] \leq G^\alpha$, $\forall c \in [C]$, $x \in \mathbb{R}^d$, where $\xi$ is the noise from the minibatch.*

Assumption 5 implies that the noise can have unbounded second moments when $\alpha < 2$, which is much weaker compared to Assumption 2. This assumption can be satisfied by a family of noises including the Pareto distribution (Arnold, 2014) and $\alpha$-stable Levy distribution (Nolan, 2012), both of which have unbounded variances[ab: cite]. Heavy-tailed noises have detrimental effects on most of existing optimization theories, while, at the same time, being prevalent in FL due to data heterogeneity, i.e., non-i.i.d. client data distributions. This phenomenon has been shown in (Charles et al., 2021), and Assumption 5 has been adopted in existing theoretical analysis (Zhang et al., 2020; Yang et al., 2022). Clipping-based methods (Koloskova et al., 2023), a mainstream approach to handle exceedingly large gradient norms, use adaptive learning rates to normalize the gradient. These methods have empirically demonstrated the capability under heavy-tailed scenarios and are also proven to have optimal convergence rates (Zhang et al., 2020; Liu et al., 2022). [ab: in non-FL settings?][lu: In both FL and non-FL settings]

Our goal is to apply the sketching techniques to the clipping-based adaptive methods. This is indeed a challenging task. As we have already shown in Section 2, random sketching introduces a significant amount of noise to the client updates. It is unknown whether these noises additionally introduced by sketching will affect the behavior of clipping methods, given that the intrinsic noises are already heavy-tailed due to data heterogeneity.

To address this open question, we propose the Sketched Adaptive Clipped Federated Learning (SACFL) in Algorithm 1 which calls Algorithm 3. In each round, besides sketching the local updates, the client directly sends the $L_2$-norm of the update to the server. The $L_2$-norm can be viewed as a global second moment specific to clipping methods. Notably, the $L_2$-norm is a scalar value and does not require any compression. Upon receiving the running moments $\bar{m}_t^c, \bar{v}_t^c$ from clients, the server averages the sketched local updates and the $L_2$-norms respectively, and then updates the global model by $x_t = x_{t-1} - \kappa \min\{\frac{\tau}{\bar{v}_t}, 1\} \texttt{desk}(\bar{m}_t)$, where $\kappa$ is the learning rate and $\tau$ is a horizon-dependent clipping threshold. When the averaged gradient norm exceeds $\tau$, i.e., when the gradient is in the heavy-tailed regime, clipping is enabled to downscale the gradient. Otherwise, the recovered gradients are directly used to update the global model.

## 3.2 Convergence Analysis

Next, we state the convergence guarantees of SACFL under Assumption 5. All technical proofs for this section are in Appendix D. We start with the descent lemma for clipping methods.

**Lemma 3.1.** *If the sketching dimension $b$ satisfies $b \geq 4\log^3(d/\delta)$, taking expectation over the stochasticity of gradients yields, with probability $1 - \Theta(\delta)$,*

$$\mathbb{E}[\mathcal{L}(x_{t+1})] - \mathcal{L}(x_t) + \frac{\kappa\eta K}{4}\|\nabla\mathcal{L}(x_t)\|^2$$

$$\leq \frac{\kappa\eta K}{2}\|\nabla\mathcal{L}(x_t) - \frac{1}{K}\frac{1}{C}\sum_{c=1}^{C}\mathbb{E}[\tilde{\Delta}_t^c]\|^2 + \frac{\kappa^2\eta^2}{2}\mathbb{E}[(\frac{1}{C}R^\top R\sum_{c=1}^{C}\tilde{\Delta}_t^c)^\top H_\mathcal{L}(\hat{x}_t)(\frac{1}{C}R^\top R\sum_{c=1}^{C}\tilde{\Delta}_t^c)],$$

*where $\tilde{\Delta}_t^c = \min\{1, \frac{\tau}{\frac{1}{C}\sum_{c=1}^{C}\|\Delta_t^c\|}\}\Delta_t^c$, and $\Delta_t^c = \sum_{k=1}^{K} g_{t,k}^c$, $H_\mathcal{L}(\hat{x}_t)$ is the loss Hessian at some $\hat{x}_t$ within the element-wise interval of $[x_t, x_{t+1}]$.*

Intuitively, the first-order terms are barely affected by the heavy-tailed noise since we assume $\alpha > 1$ and these terms do not involve the potentially-unbounded second moments, although special attention for the first-order terms is necessary to achieve the optimal convergence rate, which are deferred to the appendix. Next, we show how to deal with the second-order term. With probability $1 - \Theta(\delta)$,

$$\mathbb{E}[(\frac{1}{C}R^\top R\sum_{i=1}^{C}\tilde{\Delta}_t^c)^\top \hat{H}_\mathcal{L}(\frac{1}{C}R^\top R\sum_{i=1}^{C}\tilde{\Delta}_t^c)] = \mathbb{E}[\sum_{j=1}^{d}\lambda_j\langle\frac{1}{C}R^\top R\sum_{i=1}^{C}\tilde{\Delta}_t^c, v_j\rangle^2]$$

$$\leq \mathbb{E}\left[\sum_{j=1}^{d}\lambda_j 1_{\lambda_j\geq 0}\left(\frac{\tau M}{\frac{1}{C}\sum_{c=1}^{C}\|\Delta_t^c\|}\frac{1}{C}\sum_{c=1}^{C}\|\Delta_t^c\|\right)^{2-\alpha}\left(\frac{M}{C}\sum_{i=1}^{C}\|\Delta_t^c\|\right)^{\alpha}\right] \leq M\sum_{j=1}^{d}\lambda_j 1_{\lambda_j\geq 0}K^2\tau^{2-\alpha}G^\alpha,$$

where $M := (1 + \frac{\log^{1.5}(d/\delta)}{\sqrt{b}})$. The first equality follows by using the same eigen-decomposition as in the previous section where $\{\lambda_j, v_j\}$ are the eigenpairs of $H_\mathcal{L}(\hat{x}_t)$ and the second order term can be reduced to a squared inner product term. The primary trick thereafter (in the first inequality) is to split the inner product terms into two parts, which can be handled by the two-sided adaptive learning rates respectively. By applying the bounded second moment of random sketching, we find the first part with order $2 - \alpha$ is contained in a $(1 + \frac{\log^{1.5}(d/\delta)}{\sqrt{b}})\tau-$ball with high probability, and the second part with order $\alpha$ is bounded by applying Assumption 5.

**Remark 3.1.** The bound is high-probability w.r.t the randomness of sketching functions, while the expectation is over other randomness, including local stochastic noises and the heavy-tailed noises.

Finally, we can derive the convergence rate for SACFL by combining the analysis.

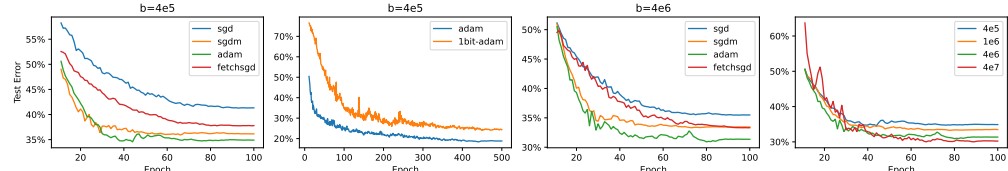

Figure 1: Test Error on CIFAR-10 with ResNet of 42M parameters. The plot starts from the 10th epoch for better demonstration. Optimizers: ADA_OPT $\in$ {SGD, SGDm, Adam}, FetchSGD and 1bit-Adam with sketch size $b \in \{4e5, 1e6, 4e6\}$; Rightmost: ADA_OPT is Adam. The legend 4e7 represents training in the ambient dimension without sketching. Adam optimizer consistently outperforms other optimizers. Larger sketch sizes improves the convergence rate and test errors.

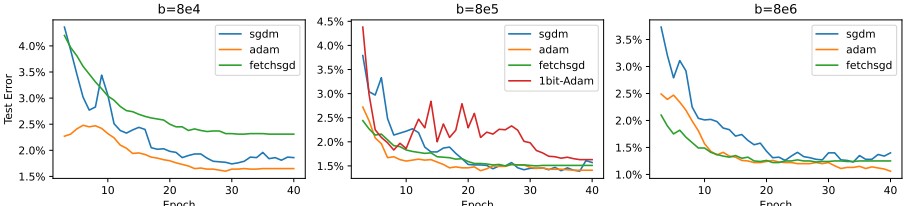

Figure 2: Test Error on CIFAR-10. We finetune a ViT-base model (with 86M parameters) from the pretrained backbone checkpoint (Dosovitskiy et al., 2020). SGDm, Adam, FetchSGD are compared under sketch size $b \in \{8e4, 8e5, 8e6\}$. 1Bit-Adam has comparable compression rates with $b = 8e5$. Sketched Adam optimizer consistently outperforms other communication-efficient algorithms.

**Theorem 3.2.** *If the sketch size $b$ satisfies $b \geq 4\log^3(dT/\delta)$, then under Assumption 4 and 5, the sequence $\{x_t\}$ generated by Algorithm 1 (SACFL) satisfies:*

$$\frac{1}{T}\sum_{t=1}^{T}\mathbb{E}[\|\nabla\mathcal{L}(x_t)\|]^2 \leq \frac{4(\mathcal{L}(x_1) - \mathcal{L}(x_T))}{\kappa\eta KT} + 3K^2(L^2\eta^2G^2 + G^{2\alpha}\tau^{-2(\alpha-1)} + L\eta G^{1+\alpha}\tau^{1-\alpha})$$

$$+ 3\hat{L}\kappa\eta(KG^\alpha\tau^{2-\alpha}), \ \ w.p. \ 1 - \Theta(\delta)$$

*With a proper choice of hyper-parameters with $\kappa = K^{\frac{3\alpha-6}{3\alpha-2}}T^{-\frac{1}{3\alpha-2}}$, $\eta = T^{\frac{1-\alpha}{3\alpha-2}}K^{\frac{4-4\alpha}{3\alpha-2}}$ and $\tau = (K^4T)^{\frac{1}{3\alpha-2}}$, we achieve $\frac{1}{T}\sum_{t=1}^{T}\mathbb{E}[\|\nabla\mathcal{L}(x_t)\|^2] \leq O(T^{\frac{2-2\alpha}{3\alpha-2}}K^{\frac{4-2\alpha}{3\alpha-2}})$, $w.p. \ 1 - \Theta(\delta)$.*

**Remark 3.2.** The convergence rate depends on the noise level $\alpha$. When $\alpha = 2$, i.e. the bounded variance case, the convergence rate is $O(1/\sqrt{T})$, which matches the rate in Theorem 2.1. We also claim that the iteration complexity matches the optimal bound in the heavy-tailed case (Yang et al., 2022; Zhang et al., 2020). □

## 4 EMPIRICAL STUDIES

In this section, we instantiate the algorithmic framework of SAFL in Section 2 and SACFL in Section 3 to demonstrate the effect of sketching in different settings.

**Experimental Configurations.** We adopt three distinct experimental settings, from vision to language tasks, and in finetuning and training-from-scratch regimes. For the vision task, we train a ResNet101 (Wu & He, 2018) with a total of 42M parameters from scratch and finetune a ViT-Base (Dosovitskiy et al., 2020) with 86M parameters on the CIFAR-10 dataset (Krizhevsky et al., 2009). For the language task, we adopt SST2, a text classification task, from the GLUE benchmark (Wang et al., 2018). We train a BERT model (Devlin, 2018) which has 100M parameters. The client optimizer is mini-batch SGD. At each round, the client trains one single epoch (iterate over the client dataset). For other hyperparameters used in the training process, please refer to the appendix.

**Sharp-Decaying Hessian Eigenspectrum**. As a key technical cornerstone of the theory, Assumption 4 states that the Hessian matrix has a sharp-decaying eigenspectrum. While this assumption has been repeatedly validated in the previous works, it's unknown if the property holds in the context of federated deep learning. We show an affirmative verification in Fig. 6 in the Appendix.

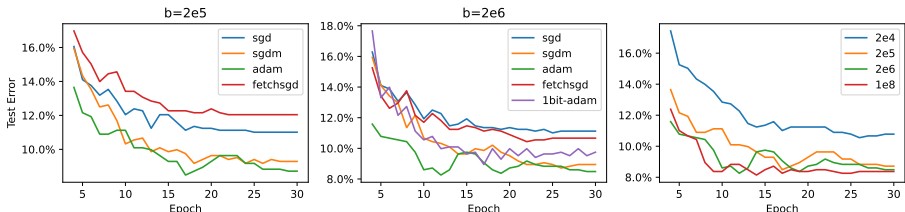

Figure 3: Test Error on SST2 (GLUE) with BERT of 100M parameters. Left: sketch size $b = 2e5$; Middle: $b = 2e6$; Right: ADA_OPT is Adam, with sketch size $b \in \{2e4, 2e5, 2e6\}$. The legend $1e8$ represents training in the ambient dimension without sketching. Adam achieves faster convergence and lower test errors across different sketch sizes. Larger sketch sizes mainly improves the convergence rate and achieve comparable test errors at the end of training.

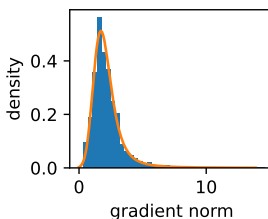 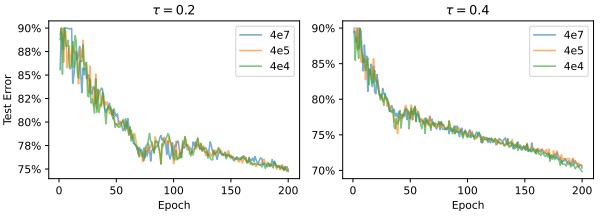

(a) Heavy tailed gradient norms.       (b) Test error under sketched clipping.

Figure 4: Sketched Clipping Methods on CIFAR10 training ResNet (40M params). (a) histogram of local gradient norms, which satisfies an $\alpha$-Levy distribution with $\alpha \approx 1.5$ (the orange curve). (b) trajectory of test errors under sketch sizes $b \in \{4e4, 4e5\}$. $4e7$ means training without sketching. Left: $\tau = 0.2$; Right: $\tau = 0.4$. With the same $\tau$, trajectories of distinct sketch sizes overlap.

**Sketched Adaptive FL.** We adopt Adam as the base adaptive optimizer at the server side, and make comparison with the sketched non-adaptive optimizers SGD, SGDm (SGD with momentum). We also involve the state-of-the-art communication-efficient algorithms FetchSGD (Rothchild et al., 2020) and 1bit-Adam (Tang et al., 2021), which are based on biased sparsification and quantization respectively. In the i.i.d client setting, the data are uniformly distributed over 5 clients. Fig. 1 depicts the test errors on CIFAR-10 when training ResNet(40M) with sketch sizes $b \in \{4e5, 1e6, 4e6\}$. We can see for all sketch sizes, our sketched Adam consistently outperforms other optimizers in the convergence rate and the test error. The compression rate of 1bit-Adam is fixed at 97%, which is comparable with the compression rate 99% achieved at $b = 4e5$. 1bit-Adam is plotted separately because it takes remarkably longer to converge. More interestingly, even with distinct sketch sizes, the iterations needed for convergence in Adam are almost the same. The test performance degrades slightly with smaller sketch sizes. This is anticipated and totally acceptable considering that the communication cost has been drastically reduced. With the same budget of communication bits, using a lower compression rate facilitates larger model training, which has the potential in better generalization performance. For results on extremely large compression rates, please refer to Appendix E.

We also present results on finetuning a ViT-Base model (80M) in Fig. 2. The sketch size $b \in \{8e4, 8e5, 8e6\}$. We see, in the finetuning regime, the sketched Adam optimizer also achieves competitive performance with the baseline methods. Similar phenomenon is observed in the language task. Fig. 3 shows the test errors of training SST2 with BERT (100M). The sketch sizes are selected from $\{2e4, 2e5, 2e6, 1e8\}$. We observe sketched Adam converges faster and achieves a slightly better test performance than other optimizers. Note that the sketch size of $2e4$ is tiny, given that the ambient dimension is 100M. It is quite thrilling that using an extremely high compression rate (99.98%), the model can still achieve comparable performance as trained in the ambient dimension.

**Sketched Clipping Method.** Next, we study the performance of the sketched clipping methods. In Section 3, we claim that SACFL excels in the heavy-tailed regime even when interfered with the noise from random sketching. To show empirical evidence, we first build an (extremely pessimistic) environment of heavy-tailed noise on the CIFAR-10 dataset. Specifically, the data categories are extremely imbalanced among 80 clients. Each client has 4 distinct majority classes which occupy 80% of its entire client dataset, while the remaining data samples are the minority categories. In this data heterogeneous setting, the features are hardly learned. We run sketched clipping methods in this environment and fit the local gradient norms with an $\alpha$-stable Levy distribution in Fig. 4(a). We select the clipping threshold $\tau$ in $\{0.2, 0.4\}$ such that the clipping operator is in effect in most rounds.

Fig. 4(b) depicts the test errors in the first 200 epochs under distinct sketch sizes, where we observe the trajectories significantly overlap. Hence, we can verify that the sketching operator has minor effects on the clipping method, while providing the benefits of lower communication costs.[ab: Need a sentence on why the test errors are large] For completeness, we present results on the BERT model trained with SST2 dataset in Fig. 8 in Appendix, where we can also observe the sketched clipping method preserves the original convergence guarantees.

## 5 CONCLUSION

In this paper, we investigated sketched adaptive methods for FL. While the motivation behind combining sketching and adaptive methods for FL is clear, there is limited understanding on its empirical success due to the inherent technical challenges. We consider both mild-noise and heavy-tailed noise settings, propose corresponding adaptive algorithms for each, and show highly promising theoretical and empirical results. Inspired by the recently observations on heterogeneity in weights across neural network layers (Zhang et al., 2024), an important future direction is to independently sketch layer-wise gradients, rather than sketching the concatenated gradient vectors. We believe our novel work can form the basis for future advances on the theme.

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

---

**Algorithm 3** `ADA_OPT` (AdaClip)

**Input:** iterate $x_{t-1}$, sketched updates $\bar{m}_t, \bar{v}_t$
**Parameter:** Learning rate $\kappa$, clipping threshold $\tau$
**Output:** Updated parameters $x_t$
Update parameters: $x_t = x_{t-1} - \kappa \min\{\frac{\tau}{\bar{v}_t}, 1\}\mathtt{desk}(\bar{m}_t)$.

---

**Algorithm 4** `ADA_OPT` (Adam)

**Input:** iterate $x_{t-1}$, moments $m_{t-1}, v_{t-1}, \hat{v}_{t-1}$, sketched updates $\bar{m}_t, \bar{v}_t$
**Parameter:** Learning rate $\kappa$, $\beta_1$, $\beta_2$, Small constant $\epsilon$
**Output:** Updated parameters $x_t$, and moments $m_t, v_t, \hat{v}_t$
Update first moment estimate: $m_t = \beta_1 \cdot m_{t-1} + (1 - \beta_1) \cdot \mathtt{desk}(\bar{m}_t)$;
Update second moment estimate: $v_t = \beta_2 \cdot v_{t-1} + (1 - \beta_2) \cdot \mathtt{desk}(\bar{v}_t)$;
Bias Correction: $\hat{m}_t = m_t/(1 - \beta_1^t)$; $\hat{v}_t = v_t/(1 - \beta_2^t)$;
Update parameters: $x_t = x_{t-1} - \frac{\kappa}{\sqrt{\hat{v}_t} + \epsilon} \cdot \hat{m}_t$.

---

## A    RELATED WORKS

**Adaptive Learning Rates.** Adaptive learning rates have long been studied. Adagrad is first proposed in (Duchi et al., 2011) in aim of utilizing sparsity in stochastic gradients. Subsequent works, e.g. Adam (Kingma & Ba, 2014) and AMSGrad (Reddi et al., 2019) have become the mainstream optimizers used in machine learning because of their superior empirical performance. These methods use implicit learning rates adaptive to the current iterate in the training process. In many cases, adaptive methods have been shown to converge faster than SGD, and with better generalization as well (Reddi et al., 2019).

**Gradient Compression.** To alleviate communcation overhead in federated learning, a promising direction is to compress the message between clients and the server. The mainstream gradient compression techniques include quantization (Alistarh et al., 2017; Chen et al., 2023; Reisizadeh et al., 2020; Liu et al., 2023a), sparsification (Alistarh et al., 2018; Wu et al., 2018; Rothchild et al., 2020) and sketching. Quantization methods reduce the overhead in storing every element of the parameters, and hence still takes $O(d)$ bits per round. Sparsification methods, e.g. Tok-K, random sparsification, increases sparsity in the gradient so that the cost is proportional to the number of non-zero elements in the sparsified gradient. Sketching techniques adopts a random sketching function to project a high-dimension vector to a low-dim subspace. The technique is promising and has been widely used in least-square regression (Tang et al., 2017), second-order optimization (Pilanci & Wainwright, 2017), and memory-efficient learning (Feinberg et al., 2024).

**Noise in Learning.** There has been literatures discussing the noise in neural network training. In our work, we are also dealing with the noise from various sources. High-probability bounds are indeed quite limited, as the mainstream of analysis of the optimization methods are over expectation. The lighted-tailed noise assumption is proposed by (Rakhlin et al., 2011) in the strongly-convex settings, which is subsequently improved by (Harvey et al., 2019). More recently, the communities find the heavy-tailed phenomenon are prevalent in general machine learning tasks (Simsekli et al., 2019; Reddi et al., 2020). It is also observed in federated learning settings when the data is heterogeneous across clients (Yang et al., 2022).

## B    LEMMA FOR RANDOM SKETCHING

For completeness, we provide the following lemmas that give high probability bounds on the inner products.

**Lemma B.1.** *(SRHT)[Same as Lemma D.23 (Song et al., 2023)] Let $R \in \mathbb{R}^{b \times d}$ denote a subsample randomized Hadamard transform or AMS sketching matrix. Then for any fixed vector $h \in \mathbb{R}$ and any fixed vector $g \in \mathbb{R}$ the following properties hold:*

$$\mathbb{P}\left[|\langle g^\top R^\top R h - g^\top h| \geq \frac{\log^{1.5}(d/\delta)}{\sqrt{b}}\|g\|_2\|h\|_2\right] \leq \Theta(\delta).$$

---

**Algorithm 5** Sketched Adaptive Federated Learning (SAFL)

---

**Input:** Learning rate $\eta$, initial parameters $x_0$, adaptive optimizer ADA_OPT
**Output:** Updated parameters $x_T$
Initialize server moments: $m_0 = 0$, $v_0 = 0$, $\hat{v}_0 = 0$, client initial parameters: $x_{0,0}^c = x_0$, client moments: $m_0^c = 0, v_0^c = 0, \hat{v}_0^c = 0, \forall c \in [C]$;
**for** $t = 1, 2, \ldots, T$ **do**
  **Client Updates:**
  **for** $c = 1, 2, \ldots, C$ **do**
    Client model synchronization:
$$x_{t,0}^c, m_t^c, v_t^c, \hat{v}_t^c = \text{ADA\_OPT}(x_{t-1,0}^c, m_{t-1}^c, v_{t-1}^c, \hat{v}_{t-1}^c, \bar{m}_t, \bar{v}_t)$$
    **for** $k = 1, 2, \ldots, K$ **do**
      Compute stochastic gradient $g_{t,k-1}^c$ with respect to the parameters $x_{t,k-1}^c$;
      Perform a single gradient step: $x_{t,k}^c = x_{t,k-1}^c - \eta_t g_{t,k-1}^c$;
    **end**
    Sketch (compress) the parameter updates:
$$\bar{m}_t^c = \text{sk}(x_{t,0}^c - x_{t,K}^c);$$
  **end**
  **Server Updates:**
  Average sketched client updates, second moment as average of elementwise square and send back to clients
$$\bar{m}_t = \frac{1}{C}\sum_{c=1}^{C} \bar{m}_t^c; \qquad \bar{v}_t = \frac{1}{C}\sum_{c=1}^{C}(\bar{m}_t^c)^2;$$
  Update paramters and moments: $x_t, m_t, v_t, \hat{v}_t = \text{ADA\_OPT}(x_{t-1}, m_{t-1}, v_{t-1}, \hat{v}_{t-1}, \bar{m}_t, \bar{v}_t)$.
**end**

---

**Lemma B.2.** *(Gaussian)[Same as Lemma D.24 (Song et al., 2023)] Let $R \in \mathbb{R}^{b \times d}$ denote a random Gaussian matrix. Then for any fixed vector $h \in \mathbb{R}$ and any fixed vector $g \in \mathbb{R}$ the following properties hold:*

$$\mathbb{P}\left[|\langle g^\top R^\top Rh - g^\top h| \geq \frac{\log^{1.5}(d/\delta)}{\sqrt{b}}\|g\|_2\|h\|_2\right] \leq \Theta(\delta).$$

**Lemma B.3.** *(Count-Sketch)[Same as Lemma D.25 (Song et al., 2023)] Let $R \in \mathbb{R}^{b \times d}$ denote a count-sketch matrix. Then for any fixed vector $h \in \mathbb{R}$ and any fixed vector $g \in \mathbb{R}$ the following properties hold:*

$$\mathbb{P}\left[|\langle g^\top R^\top Rh - g^\top h| \geq \log(1/\delta)\|g\|_2\|h\|_2\right] \leq \Theta(\delta).$$

## C  PROOF OF THEOREM 2.1

### C.1  PROOF OF LEMMA 2.2

Let
$$z_t = x_t + \frac{\beta_1}{1-\beta_1}(x_t - x_{t-1}) = \frac{1}{1-\beta_1}x_t - \frac{\beta_1}{1-\beta_1}x_{t-1}.$$

---

**Algorithm 6** Sketched Adaptive Clipped Federated Learning (SACFL)

---

**Input:** Learning rate $\kappa$, $\eta$, initial parameters $x_0$, clipping threshold $\tau$.
**Output:** Updated parameters $x_T$
Initialize client initial parameters: $x_{0,0}^c = x_0, \forall c \in [C]$;
  **for** $t = 1, 2, \ldots, T$ **do**
    **for** $c = 1, 2, \ldots, C$ **do**
      De-sketch the updates: $x_{t,0}^c = x_{t-1} - \kappa \min\{\frac{\tau}{\bar{v}_t}, 1\} \texttt{desk}(\bar{m}_t)$;
      **for** $k = 1, 2, \ldots, K$ **do**
        Compute stochastic gradient $g_{t,k-1}^c$ with respect to the parameters $x_{t,k-1}^c$;
        Perform a single gradient step: $x_{t,k}^c = x_{t,k-1}^c - \eta g_{t,k}^c$;
      **end**
      Sketch the parameter updates:
$$\bar{m}_t^c = \texttt{sk}(x_{t,0}^c - x_{t,K}^c); \qquad \bar{v}_t^c = \|x_{t,K}^c - x_{t,0}^c\|;$$
    **end**
    Average client updates and send back the averages:
$$\bar{m}_t = \frac{1}{C} \sum_{c=1}^{C} \bar{m}_t^c; \qquad \bar{v}_t = \frac{1}{C} \sum_{c=1}^{C} \bar{v}_t^c;$$
    Update paramters and statistics: $x_t = x_{t-1} - \kappa \min\{\frac{\tau}{\bar{v}_t}, 1\} \texttt{desk}(\bar{m}_t)$.
  **end**

---

Then, the update on $z_t$ can be expressed as

$$z_{t+1} - z_t = \frac{1}{1 - \beta_1}(x_{t+1} - x_t) - \frac{\beta_1}{1 - \beta_1}(x_t - x_{t-1})$$

$$= -\frac{1}{1 - \beta_1} \kappa \hat{V}_t^{-1/2} \cdot m_t + \frac{\beta_1}{1 - \beta_1} \kappa \hat{V_{t-1}}^{-1/2} \cdot m_{t-1}$$

$$= -\frac{1}{1 - \beta_1} \kappa \hat{V}_t^{-1/2} \cdot (\beta_1 m_{t-1} + (1 - \beta_1) \cdot R_t^\top \bar{m}_t) + \frac{\beta_1}{1 - \beta_1} \kappa \hat{V}_{t-1}^{-1/2} \cdot m_{t-1}$$

$$= \frac{\beta_1}{1 - \beta_1} \left( \kappa \hat{V}_{t-1}^{-1/2} - \kappa \hat{V}_t^{-1/2} \right) m_{t-1} - \frac{\kappa}{C} \hat{V}_t^{-1/2} R_t^\top \sum_{c=1}^{C} \bar{m}_t^c$$

$$= \frac{\beta_1}{1 - \beta_1} \left( \kappa \hat{V}_{t-1}^{-1/2} - \kappa \hat{V}_t^{-1/2} \right) m_{t-1} - \frac{\kappa}{C} \hat{V}_t^{-1/2} R_t^\top \sum_{c=1}^{C} R_t(x_{t,0}^c - x_{t,K}^c)$$

$$= \frac{\beta_1}{1 - \beta_1} \left( \kappa \hat{V}_{t-1}^{-1/2} - \kappa \hat{V}_t^{-1/2} \right) m_{t-1} - \frac{\kappa \eta}{C} \hat{V}_t^{-1/2} \sum_{c=1}^{C} \sum_{k=1}^{K} R_t^\top R_t g_{t,k}^c$$

By Taylor expansion, we have

$$\mathcal{L}(z_{t+1}) = \mathcal{L}(z_t) + \nabla \mathcal{L}(z_t)^\top (z_{t+1} - z_t) + \frac{1}{2}(z_{t+1} - z_t)^\top \hat{H}_{\mathcal{L}}(z_{t+1} - z_t)$$

$$= \mathcal{L}(z_t) + \nabla \mathcal{L}(x_t)^\top (z_{t+1} - z_t) + (\nabla \mathcal{L}(z_t) - \nabla \mathcal{L}(x_t))^\top (z_{t+1} - z_t) + \frac{1}{2}(z_{t+1} - z_t)^\top \hat{H}_{\mathcal{L}}(z_{t+1} - z_t).$$

Bounding the first-order term

$$\nabla\mathcal{L}(x_t)^\top(z_{t+1} - z_t)$$

$$=\nabla\mathcal{L}(x_t)^\top\left(\frac{\beta_1}{1-\beta_1}\left(\kappa\hat{V}_{t-1}^{-1/2} - \kappa\hat{V}_t^{-1/2}\right)m_{t-1} - \frac{\kappa\eta}{C}\hat{V}_t^{-1/2}\sum_{c=1}^C\sum_{k=1}^K R_t^\top R_t g_{t,k}^c\right)$$

$$\leq\frac{\beta_1}{1-\beta_1}\|\nabla\mathcal{L}(x_t)\|_\infty(\|\kappa\hat{V}_{t-1}^{-1/2}\|_{1,1} - \|\kappa\hat{V}_t^{-1/2}\|_{1,1})\|m_{t-1}\|_\infty$$

$$-\frac{\eta}{C}\nabla\mathcal{L}(x_t)^\top(\kappa\hat{V}_t^{-1/2} - \kappa\hat{V}_{t-1}^{-1/2})\sum_{c=1}^C\sum_{k=1}^K R_t^\top R_t g_{t,k}^c - \frac{\kappa\eta}{C}\nabla\mathcal{L}(x_t)^\top\hat{V}_{t-1}^{-1/2}\sum_{c=1}^C\sum_{k=1}^K R_t^\top R_t g_{t,k}^c$$

$$\leq\left(\frac{\beta_1}{1-\beta_1}\|m_{t-1}\|_\infty + \frac{\eta}{C}\|\sum_{c=1}^C\sum_{k=1}^K R_t^\top R_t g_{t,k}^c\|_\infty\right)\|\nabla\mathcal{L}(x_t)\|_\infty(\|\kappa\hat{V}_{t-1}^{-1/2}\|_{1,1} - \|\kappa\hat{V}_t^{-1/2}\|_{1,1})$$

$$-\frac{\kappa\eta}{C}\sum_{c=1}^C\sum_{k=1}^K\nabla\mathcal{L}(x_t)^\top\hat{V}_{t-1}^{-1/2}R_t^\top R_t g_{t,k}^c.$$

The quadratic terms can be written as

$$(\nabla\mathcal{L}(z_t) - \nabla\mathcal{L}(x_t))^\top(z_{t+1} - z_t) = (z_t - x_t)^\top\hat{H}_\mathcal{L}\left(\frac{1}{1-\beta_1}(x_{t+1} - x_t) - \frac{\beta_1}{1-\beta_1}(x_t - x_{t-1})\right),$$

where $\hat{H}_\mathcal{L}$ is a second-order Taylor remainder. So the quadratic term can be further seen as a quadratic form over $z_{t+1} - z_t$ and $z_t - x_t$, denote as $\mathcal{Q}(z_{t+1} - z_t, z_t - x_t)$. For the same reason, the term $\frac{1}{2}(z_{t+1} - z_t)^\top\hat{H}_\mathcal{L}(z_{t+1} - z_t)$ can also be written into a quadratic form $\mathcal{Q}(z_{t+1} - z_t, z_{t+1} - z_t)$. Putting the two terms together yields a quadratic form of $\mathcal{Q}(z_{t+1} - z_t, z_t - x_t)$.

## C.2 PROOF OF LEMMA C.1 (GENERALIZED VERSION OF LEMMA 2.3)

*Proof.* We can prove by induction. For $t = 0$, since $m_0 = 0$, the inequality holds. Suppose we have for $h \in \mathbb{R}^d$, s.t. $\|h\| \leq H$, with probability $1 - \Theta((t-1)\delta)$,

$$|m_{t-1}^\top h| \leq (1 + \frac{\log^{1.5}(CKd/\delta)}{\sqrt{b}})G$$

Then by the update rule,

$$|m_t^\top h| = |(\beta_1 \cdot m_{t-1} + (1-\beta_1)\cdot\frac{\eta}{C}\sum_{c=1}^C\sum_{k=1}^K R_t^\top R_t g_{t,k}^c)^\top h|$$

$$\leq \beta_1|m_{t-1}^\top h| + \frac{(1-\beta_1)\eta}{C}\sum_{c=1}^C\sum_{k=1}^K|\langle R_t^\top R_t g_{t,k}^c, h\rangle|$$

$$\leq \beta_1|m_{t-1}^\top h| + (1-\beta_1)(1 + \frac{\log^{1.5}(CKd/\delta)}{\sqrt{b}})\eta\sum_{k=1}^K\|g_{t,k}^c\|_2\|h\|_2$$

$$\leq (1 + \frac{\log^{1.5}(CKd/\delta)}{\sqrt{b}})\eta KGH, \quad w.p.\ 1 - \Theta(t\delta).$$

Let $h = \hat{V}_t^{-1/2}v_i$. Then $\|h\|_2 \leq 1/\epsilon$. We have

$$|(\hat{V}_t^{-1/2}m_t)^\top v_i| \leq (1 + \frac{\log^{1.5}(CKd/\delta)}{\sqrt{b}})\eta KG/\epsilon$$

$\square$

### C.3 PROOF OF LEMMA 2.4

We first prove the element-wise lower bound of the diagonal matrix $\hat{V}_{t-1}^{-1/2}$. Denote $(\hat{V}_{t-1}^{-1/2})_i$ as the $i$-th element on the diagonal of $\hat{V}_{t-1}^{-1/2}$. By the update rule,

$$(\hat{V}_{t-1}^{-1/2})_i \geq (\max_{t-1}(\sqrt{v_{t,i}}) + \epsilon)^{-1} \geq (\sqrt{1 + \frac{\log^{1.5}(CKtd/\delta)}{\sqrt{b}}}\eta KG + \epsilon)^{-1}, \;\; w.p. \; 1 - \Theta(\delta)$$

where the last inequality follows by letting $h$ as a one-hot vector $h_i$ in Lemma B.1, observing that the elements can be transformed to an inner product form $v_{t,i} = v_t^\top h_i$. Then the scaled gradient norm can be lower bounded as

$$\nabla\mathcal{L}(x_t)^\top \hat{V}_{t-1}^{-1/2}\nabla\mathcal{L}(x_t) \geq \min_i(\hat{V}_{t-1}^{-1/2})_i \sum_{i=1}^d [\nabla\mathcal{L}(x_t)]_i^2$$

$$\geq (\sqrt{1 + \frac{\log^{1.5}(CKtd/\delta)}{\sqrt{b}}}\eta KG + \epsilon)^{-1}\|\nabla\mathcal{L}(x_t)\|^2, \;\; w.p. \; 1 - \Theta(d\delta)$$

which completes the proof by applying union bounded on the dimension $d$.

### C.4 PROOF OF LEMMA 2.5

Since the noise is zero-centered, we view the random process of

$$\{Y_t = \sum_{\tau=1}^t \frac{1}{C}\sum_{c=1}^C\sum_{k=1}^K \nabla\mathcal{L}(x_\tau)^\top \hat{V}_{\tau-1}^{-1/2}(R_\tau^\top R_\tau g_{\tau,k}^c - g_{\tau,k}^c)\}_{t=1}^T$$

as a martingale. The difference of $|Y_{t+1} - Y_t|$ is bounded with high probability

$$|Y_{t+1} - Y_t| = |\nabla\mathcal{L}(x_t)^\top \hat{V}_{t-1}^{-1/2}(R_t^\top R_t g_{t,k}^c - g_{t,k}^c)| \leq \frac{\log^{1.5}(d/\delta)}{\sqrt{b}}G\|\hat{V}_t^{-1/2}\nabla\mathcal{L}(x_t)\|_2, \;\; w.p. \; 1 - \Theta(\delta)$$

Then by Azuma's inequality,

$$\mathbb{P}(|Y_T| \geq \nu\sqrt{\sum_{t=1}^T \left(\frac{\log^{1.5}(d/\delta)}{\sqrt{b}}G\|\hat{V}_t^{-1/2}\nabla\mathcal{L}(x_t)\|_2\right)^2}) = O(\exp(-\Omega(\nu^2))) + T\delta \qquad (1)$$

Note that the original Azuma's is conditioned on a uniform bound of the difference term, while our bound here is of high probability. Hence, we need another union bound. A similar bound can be achieved for the sub-Gaussian noise in stochastic gradient. Let

$$Z_t = \sum_{\tau=1}^t \frac{1}{C}\sum_{c=1}^C\sum_{k=1}^K \nabla\mathcal{L}(x_\tau)^\top \hat{V}_{\tau-1}^{-1/2}(g_{\tau,k}^c - \nabla\mathcal{L}^c(x_{t,k}^c)).$$

Then

$$\mathbb{P}(|Z_T| \geq \nu\sqrt{\sum_{t=1}^T \frac{\sigma^2}{\epsilon^2}\log(\frac{2T}{\delta_g})}) = O(\exp(-\Omega(\nu^2))) + \delta_g$$

Combining the two bounds by union bound completes the proof.

## C.5 PROOF OF THEOREM 2.1

After applying Lemma 2.2. The second order quadratic forms in the descent lemma can be written as

$$(\nabla\mathcal{L}(z_t) - \nabla\mathcal{L}(x_t))^\top (z_{t+1} - z_t)$$

$$= (z_t - x_t)^\top \hat{H}_\mathcal{L} (\frac{1}{1-\beta_1}(x_{t+1} - x_t) - \frac{\beta_1}{1-\beta_1}(x_t - x_{t-1}))$$

$$= -\kappa \frac{\beta_1}{1-\beta_1} (\hat{V}_{t-1}^{-1/2} m_{t-1})^\top \hat{H}_\mathcal{L} (\frac{1}{1-\beta_1}(-\kappa \hat{V}_t^{-1/2} m_t) - \frac{\beta_1}{1-\beta_1}(-\kappa \hat{V}_{t-1}^{-1/2} m_{t-1}))$$

$$= \kappa^2 \frac{\beta_1}{(1-\beta_1)^2} (\hat{V}_{t-1}^{-1/2} m_{t-1})^\top \hat{H}_\mathcal{L} (\hat{V}_t^{-1/2} m_t) - \kappa^2 \frac{\beta_1^2}{(1-\beta_1)^2} (\hat{V}_{t-1}^{-1/2} m_{t-1})^\top \hat{H}_\mathcal{L} (\hat{V}_{t-1}^{-1/2} m_{t-1}),$$

and

$$(z_{t+1} - z_t)^\top \hat{H}_\mathcal{L} (z_{t+1} - z_t)$$

$$= (\frac{1}{1-\beta_1}(x_{t+1} - x_t) - \frac{\beta_1}{1-\beta_1}(x_t - x_{t-1}))^\top \hat{H}_\mathcal{L} (\frac{1}{1-\beta_1}(x_{t+1} - x_t) - \frac{\beta_1}{1-\beta_1}(x_t - x_{t-1}))$$

$$= \frac{1}{(1-\beta_1)^2}(x_{t+1} - x_t)^\top \hat{H}_\mathcal{L}(x_{t+1} - x_t) - \frac{2\beta_1}{(1-\beta_1)^2}(x_{t+1} - x_t)^\top \hat{H}_\mathcal{L}(x_t - x_{t-1})$$

$$+ \frac{\beta_1^2}{(1-\beta_1)^2}(x_t - x_{t-1})^\top \hat{H}_\mathcal{L}(x_t - x_{t-1}),$$

which is essentially a quadratic form defined on $\hat{V}_t^{-1/2} m_t$ and $\hat{V}_{t-1}^{-1/2} m_{t-1}$. Hence, we provide a generalized version of Lemma 2.3, as follows.

**Lemma C.1.** *With probability* $1 - \Theta(t\delta)$, *for eigenvector* $v_i$ *of the Hessian matrix,* $|(\hat{V}_t^{-1/2} m_t)^\top v_i| \le (1 + \frac{\log^{1.5}(CKd/\delta)}{\sqrt{b}})\eta KG/\epsilon.$

Note that $v_i$ can be any basis and is constant throughout the training process. Then the sum of quadratic forms is written as

$$(\nabla\mathcal{L}(z_t) - \nabla\mathcal{L}(x_t))^\top (z_{t+1} - z_t)$$

$$\le \kappa^2 \frac{\beta_1}{(1-\beta_1)^2} (\hat{V}_{t-1}^{-1/2} m_{t-1})^\top \hat{H}_\mathcal{L} (\hat{V}_t^{-1/2} m_t) - \kappa^2 \frac{\beta_1^2}{(1-\beta_1)^2} (\hat{V}_{t-1}^{-1/2} m_{t-1})^\top \hat{H}_\mathcal{L} (\hat{V}_{t-1}^{-1/2} m_{t-1}),$$

$$= \kappa^2 \frac{\beta_1}{(1-\beta_1)^2} \sum_{i=1}^d \lambda_i (\hat{V}_{t-1}^{-1/2} m_{t-1})^\top (v_i v_i^\top) \hat{V}_t^{-1/2} m_t - \kappa^2 \frac{\beta_1^2}{(1-\beta_1)^2} \sum_{i=1}^d \lambda_i (\hat{V}_{t-1}^{-1/2} m_{t-1})^\top (v_i v_i^\top) \hat{V}_{t-1}^{-1/2} m_{t-1}$$

$$\le \kappa^2 \frac{\beta_1}{(1-\beta_1)^2} \sum_{i=1}^d |\lambda_i| |(\hat{V}_{t-1}^{-1/2} m_{t-1})^\top v_i| |(\hat{V}_t^{-1/2} m_t)^\top v_i| + \kappa^2 \frac{\beta_1^2}{(1-\beta_1)^2} \sum_{i=1}^d |\lambda_i| |(\hat{V}_{t-1}^{-1/2} m_{t-1})^\top v_i|^2$$

$$\le \kappa^2 \frac{2}{(1-\beta_1)^2} \hat{L} (1 + \frac{\log^{1.5}(CKd/\delta)}{\sqrt{b}})^2 \eta^2 K^2 G^2 / \epsilon^2,$$

where the last inequality is by $\beta_1 \le 1$ and Lemma. C.1.

**First-Order Descent Term**. The first-order term in the descent lemma can be decomposed into three components, which we will handle separately.

$$\nabla\mathcal{L}(x_t)^\top \hat{V}_{t-1}^{-1/2} R_t^\top R_t g_{t,k}^c = \underbrace{\nabla\mathcal{L}(x_t)^\top \hat{V}_{t-1}^{-1/2} \nabla\mathcal{L}^c(x_t)}_{\mathcal{D}_1} + \underbrace{\nabla\mathcal{L}(x_t)^\top \hat{V}_{t-1}^{-1/2} (R_t^\top R_t g_{t,k}^c - \nabla\mathcal{L}^c(x_{t,k}^c))}_{\mathcal{D}_2}$$

$$+ \underbrace{\nabla\mathcal{L}(x_t)^\top \hat{V}_{t-1}^{-1/2} (\nabla\mathcal{L}^c(x_{t,k}^c) - \nabla\mathcal{L}^c(x_t))}_{\mathcal{D}_3}.$$

First, $\mathcal{D}_3$ can be reduced to a second-order term by smoothness over $\mathcal{L}$,

$$\nabla\mathcal{L}(x_t)^\top \hat{V}_{t-1}^{-1/2}(\nabla\mathcal{L}^c(x_{t,k}^c) - \nabla\mathcal{L}^c(x_t)) = \nabla\mathcal{L}(x_t)^\top \hat{V}_{t-1}^{-1/2}\hat{H}_L^c(x_{t,k}^c - x_t)$$

$$= -\eta\sum_{\tau=1}^{k}\nabla\mathcal{L}(x_t)^\top \hat{V}_{t-1}^{-1/2}\hat{H}_{\mathcal{L}}^c g_{t,\tau}^c$$

$$\leq \frac{1}{\epsilon}L\|\nabla L\|\sum_{\tau=1}^{k}\|g_{t,\tau}^c\| \leq \frac{1}{\epsilon}\eta LKG^2.$$

Note that this term does not involve any stochasticity with regard to random sketching, which means we can directly derive the upper bound by Cauchy-Schwartz in the last inequality.

Next observing that $\frac{1}{C}\sum_{c=1}^{C}\nabla\mathcal{L}^c(x_t) = \nabla\mathcal{L}(x_t)$, $\mathcal{D}_1$ composes a scaled squared gradient norm. Applying element-wise high probability bound on random sketching yields the lower bound for the scale. By Lemma 2.4, we can derive the lower bound for $\mathcal{D}_1$. Note that applying union bound to $\mathcal{D}_1$ does not introduce another $T$ dependence, since $\hat{v}_{t,i}$ is monotonically non-decreasing.

**Martingale for zero-centered noise.** $\mathcal{D}_2$ contains a zero-centered noise term $R_t^\top R_t g_{t,k}^c - \nabla\mathcal{L}^c(x_{t,k}^c)$, where the randomness is over $R_t$ and the mini-batch noise at round $t$. Despite $x_{t,k}^c$ has temporal dependence, the fresh noise at round $t$ is independent of the randomness in the previous iterations. Hence, the random process defined by the aggregation of these norm terms over time forms a martingale. By Lemma 2.5, we can bound this term $\mathcal{D}_2$.

Finally, putting these parts together by union bound over $[T]$ and telescoping the descent lemma leads to Theorem 2.1.

### C.6 PROOF OF COROLLARY 1

In the aysmptotic regime, with sufficiently large $T$, the term $\sqrt{1 + \frac{\log^{1.5}(CKd^2T^2/\delta)}{\sqrt{b}}}\eta KG$ approaches $\epsilon$, so the denominator on the LHS can be replaced with $2\epsilon$. Then the derivation is straightforward by just substituting $\eta = \frac{1}{\sqrt{T}K}$ into Theorem 2.1.

### C.7 PROOF OF COROLLARY 2

We first develop the convergence bound in Theorem 2.1 under the condition $b \geq \log^3(CKd^2T^2/\delta)$,

$$\left(\sqrt{2}\eta KG + \epsilon\right)^{-1}\kappa\eta K\sum_{t=1}^{T}\|\nabla\mathcal{L}(x_t)\|^2 \leq \mathcal{L}(z_1) + \frac{1}{\epsilon}\kappa\eta^2 LK^2G^2T$$

$$+ \nu\kappa\eta K\sqrt{T}(\frac{G^2}{\epsilon} + \frac{\sigma}{\epsilon}\log^{\frac{1}{2}}(\frac{2T}{\delta_g})) + \eta^2\kappa^2 T\frac{32}{(1-\beta_1)^2}\frac{\hat{L}K^2G^2}{\epsilon^2},$$

The condition on $T \leq \frac{J_1 - \sqrt{2}G}{\epsilon^2}$ is equivalent to

$$\frac{\sqrt{2}\eta KG + \epsilon}{\eta K} \leq J_1,$$

since $\eta = \frac{1}{\sqrt{T}K}$. Then scaling the coefficient on the left hand side and substituting $\frac{1}{\sqrt{T}K}$ for $\eta$, we derive

$$\frac{1}{J_1 T}\sum_{t=1}^{T}\|\nabla\mathcal{L}(x_t)\|^2 \leq \frac{\mathcal{L}(z_1)\epsilon}{\kappa T} + \frac{1}{\epsilon}\frac{LG^2}{T} + \frac{\nu}{T}(G^2 + \sigma\log^{\frac{1}{2}}(\frac{2T}{\delta_g})) + \frac{\kappa}{T}\frac{32}{(1-\beta_1)^2}\frac{\hat{L}G^2}{\epsilon},$$

### C.8 A NON-ASYMPTOTIC BOUND ON PRACTICAL LEARNING RATES

We first state a convergence bound on using practical learning rates, which decays as the optimization procedure.

**Theorem C.2.** *Suppose the sequence of iterates $\{x_t\}_{t=1}^{T}$ is generated by Algorithm 1 with a decaying learning rate $\eta_t = \frac{1}{\sqrt{t+T_0}K}$, where $T_0 = \lceil\frac{1}{1-\beta_1^2}\rceil$. Under Assumptions 1-4, for any $T$ and $\epsilon > 0$,*

*with probability $1 - \Theta(\delta) - O(\exp(-\Omega(\nu^2))) - \delta_g$,*

$$\sum_{t=1}^{T} \left( \sqrt{1 + \frac{\log^{1.5}(CKd^2T^2/\delta)}{\sqrt{b}}} \eta_t JKG + \epsilon \right)^{-1} \kappa\eta_t \|\nabla\mathcal{L}(x_t)\|^2 \leq \mathcal{L}(z_1) + \frac{1}{\epsilon}\kappa LG^2 \log T$$

$$+ \nu\kappa \log T(\frac{\log^{1.5}(CKTd/\delta)}{\sqrt{b}}\frac{G^2}{\epsilon} + \frac{\sigma}{\epsilon}\log^{\frac{1}{2}}(\frac{2T}{\delta_g})) + \kappa^2 \log T(1 + \frac{\log^{1.5}(CKdT^2/\delta)}{\sqrt{b}})^2 \frac{8}{(1-\beta_1)^2}\frac{\hat{L}G^2}{\epsilon^2},$$

*where $\delta, \delta_g$, and $\nu$ are the randomness from sketching, sub-Gaussian stochastic noise and martingales respectively, and $J$ is a constant defined in Lemma. C.3.*

Alike the analysis in the constant learning rate case, we first define auxiliary variables $z_t$

$$z_t = x_t + \frac{\beta_1}{1-\beta_1}(x_t - x_{t-1}) = \frac{1}{1-\beta_1}x_t - \frac{\beta_1}{1-\beta_1}x_{t-1}.$$

Then, the update on $z_t$ can be expressed as

$$z_{t+1} - z_t = \frac{1}{1-\beta_1}(x_{t+1} - x_t) - \frac{\beta_1}{1-\beta_1}(x_t - x_{t-1})$$

$$= \frac{\beta_1}{1-\beta_1}\left(\kappa\hat{V}_{t-1}^{-1/2} - \kappa\hat{V}_t^{-1/2}\right)m_{t-1} - \frac{\kappa\eta_t}{C}\hat{V}_t^{-1/2}\sum_{c=1}^{C}\sum_{k=1}^{K}R_t^{\top}R_t g_{t,k}^c$$

By Taylor expansion, we have

$$\mathcal{L}(z_{t+1}) = \mathcal{L}(z_t) + \nabla\mathcal{L}(z_t)^{\top}(z_{t+1} - z_t) + \frac{1}{2}(z_{t+1} - z_t)^{\top}\hat{H}_{\mathcal{L}}(z_{t+1} - z_t)$$

$$= \mathcal{L}(z_t) + \nabla\mathcal{L}(x_t)^{\top}(z_{t+1} - z_t) + (\nabla\mathcal{L}(z_t) - \nabla\mathcal{L}(x_t))^{\top}(z_{t+1} - z_t) + \frac{1}{2}(z_{t+1} - z_t)^{\top}\hat{H}_{\mathcal{L}}(z_{t+1} - z_t).$$

Bounding the first-order term

$$\nabla\mathcal{L}(x_t)^{\top}(z_{t+1} - z_t)$$

$$= \nabla\mathcal{L}(x_t)^{\top}\left(\frac{\beta_1}{1-\beta_1}\left(\kappa\hat{V}_{t-1}^{-1/2} - \kappa\hat{V}_t^{-1/2}\right)m_{t-1} - \frac{\kappa\eta_t}{C}\hat{V}_t^{-1/2}\sum_{c=1}^{C}\sum_{k=1}^{K}R_t^{\top}R_t g_{t,k}^c\right)$$

$$\leq \frac{\beta_1}{1-\beta_1}\|\nabla\mathcal{L}(x_t)\|_{\infty}(\|\kappa\hat{V}_{t-1}^{-1/2}\|_{1,1} - \|\kappa\hat{V}_t^{-1/2}\|_{1,1})\|m_{t-1}\|_{\infty}$$

$$- \frac{\eta_t}{C}\nabla\mathcal{L}(x_t)^{\top}(\kappa\hat{V}_t^{-1/2} - \kappa\hat{V}_{t-1}^{-1/2})\sum_{c=1}^{C}\sum_{k=1}^{K}R_t^{\top}R_t g_{t,k}^c - \frac{\kappa\eta_t}{C}\nabla\mathcal{L}(x_t)^{\top}\hat{V}_{t-1}^{-1/2}\sum_{c=1}^{C}\sum_{k=1}^{K}R_t^{\top}R_t g_{t,k}^c$$

$$\leq \left(\frac{\beta_1}{1-\beta_1}\|m_{t-1}\|_{\infty} + \frac{\eta_t}{C}\|\sum_{c=1}^{C}\sum_{k=1}^{K}R_t^{\top}R_t g_{t,k}^c\|_{\infty}\right)\|\nabla\mathcal{L}(x_t)\|_{\infty}(\|\kappa\hat{V}_{t-1}^{-1/2}\|_{1,1} - \|\kappa\hat{V}_t^{-1/2}\|_{1,1})$$

$$- \frac{\kappa\eta_t}{C}\sum_{c=1}^{C}\sum_{k=1}^{K}\nabla\mathcal{L}(x_t)^{\top}\hat{V}_{t-1}^{-1/2}R_t^{\top}R_t g_{t,k}^c.$$

The quadratic terms can be written as

$$(\nabla\mathcal{L}(z_t) - \nabla\mathcal{L}(x_t))^{\top}(z_{t+1} - z_t) = (z_t - x_t)^{\top}\hat{H}_{\mathcal{L}}(\frac{1}{1-\beta_1}(x_{t+1} - x_t) - \frac{\beta_1}{1-\beta_1}(x_t - x_{t-1})),$$

where $\hat{H}_{\mathcal{L}}$ is a second-order Taylor remainder.

To bound the quadratic term, the counterpart of Lemma C.1 can be stated as

**Lemma C.3.** *With learning rate $\eta_t = O(\frac{1}{\sqrt{t+T_0}})$, where $T_0 = \lceil\frac{1}{1-\beta_1^2}\rceil$. Denote $J = \frac{1-\beta_1}{\sqrt{T_0+1}}/(\frac{1}{\sqrt{T_0+1}} - \frac{\beta_1}{\sqrt{T_0}})$. Then with probability $1 - \Theta(t\delta)$,*

$$|m_{t-1}^{\top}h| \leq (1 + \frac{\log^{1.5}(CKd/\delta)}{\sqrt{b}})JKGH$$

*Proof.* For $t = 0$, since $m_0 = 0$, the inequality holds. Suppose we have for $h \in \mathbb{R}^d$, s.t. $\|h\| \leq H$, with probability $1 - \Theta((t-1)\delta)$,

$$|m_{t-1}^\top h| \leq (1 + \frac{\log^{1.5}(CKd/\delta)}{\sqrt{b}})JKGH$$

By the update rule,

$$|m_t^\top h| = |(\beta_1 \cdot m_{t-1} + (1 - \beta_1) \cdot \frac{\eta}{C} \sum_{c=1}^C \sum_{k=1}^K R_t^\top R_t g_{t,k}^c)^\top h|$$

$$\leq \beta_1 |m_{t-1}^\top h| + \frac{(1 - \beta_1)\eta}{C} \sum_{c=1}^C \sum_{k=1}^K |\langle R_t^\top R_t g_{t,k}^c, h \rangle|$$

$$\leq \beta_1 |m_{t-1}^\top h| + (1 - \beta_1)(1 + \frac{\log^{1.5}(CKd/\delta)}{\sqrt{b}})\eta_t \sum_{k=1}^K \|g_{t,k}^c\|_2 \|h\|_2$$

$$\leq (1 + \frac{\log^{1.5}(CKd/\delta)}{\sqrt{b}})\eta_t JKGH, \quad w.p. \ 1 - \Theta(t\delta).$$

$\square$

By exactly the same as in Sec. C.3, we can lower bound the scaled gradient term by

$$\nabla \mathcal{L}(x_t)^\top \hat{V}_{t-1}^{-1/2} \nabla \mathcal{L}(x_t) \geq \min_i (\hat{V}_{t-1}^{-1/2})_i \sum_{i=1}^d [\nabla \mathcal{L}(x_t)]_i^2$$

$$\geq (\sqrt{1 + \frac{\log^{1.5}(CKtd/\delta)}{\sqrt{b}}} \eta_t KG + \epsilon)^{-1} \|\nabla \mathcal{L}(x_t)\|^2, \quad w.p. \ 1 - \Theta(d\delta).$$

On the martingale of zero-centered noises, we can simply incorporate the learning rate $\eta_t$ into the martingale. Define the random process of sketching noise as

$$\{Y_t = \sum_{\tau=1}^t \frac{\eta_\tau}{C} \sum_{k=1}^K \nabla \mathcal{L}(x_\tau)^\top \hat{V}_{\tau-1}^{-1/2} (R_\tau^\top R_\tau g_{\tau,k}^c - g_{\tau,k}^c)\}_{t=1}^T$$

as a martingale. The difference of $|Y_t - Y_{t-1}|$ is bounded with high probability

$$|Y_t - Y_{t-1}| = |\frac{\eta_t}{C} \sum_{c=1}^C \sum_{k=1}^K \nabla \mathcal{L}(x_t)^\top \hat{V}_{t-1}^{-1/2} (R_t^\top R_t g_{t,k}^c - g_{t,k}^c)|$$

$$\leq \frac{\log^{1.5}(d/\delta)}{\sqrt{b}} \eta_t KG \|\hat{V}_t^{-1/2} \nabla \mathcal{L}(x_t)\|_2, \quad w.p. \ 1 - \Theta(CK\delta).$$

Then by Azuma's inequality,

$$\mathbb{P}(|Y_T| \geq \nu \sqrt{\sum_{t=1}^T \left(\frac{\log^{1.5}(d/\delta)}{\sqrt{b}} \eta_t KG \|\hat{V}_t^{-1/2} \nabla \mathcal{L}(x_t)\|_2\right)^2}) = O(\exp(-\Omega(\nu^2))) + T\delta \quad (2)$$

A similar bound can be achieved for the sub-Gaussian noise in stochastic gradient. Let

$$Z_t = \sum_{\tau=1}^t \frac{\eta_\tau}{C} \sum_{k=1}^K \nabla \mathcal{L}(x_\tau)^\top \hat{V}_{\tau-1}^{-1/2} (g_{\tau,k}^c - \nabla \mathcal{L}^c(x_{t,k}^c)).$$

Then

$$\mathbb{P}(|Z_T| \geq \nu \sqrt{\sum_{t=1}^T (\frac{\eta_t \sigma}{\epsilon})^2 \log(\frac{2T}{\delta_g})}) = O(\exp(-\Omega(\nu^2))) + \delta_g$$

Combining the two bounds by union bound completes the proof.

## D PROOF OF THEOREM 3.2

### D.1 PROOF OF LEMMA 3.1

Denote $\Delta_t^c = \sum_{k=1}^K g_{t,k}^c$, $\tilde{\Delta}_t^c = \min\{1, \frac{\tau}{\frac{1}{C}\sum_{c=1}^C \|\Delta_t^c\|}\}\Delta_t^c$. Then $x_{t+1} - x_t = -\kappa\eta R^\top R \frac{1}{C}\sum_{c=1}^C \tilde{\Delta}_t^c$.

*Proof.* Taking the expectation of randomness in stochastic gradient yields

$$\mathbb{E}[\mathcal{L}(x_{t+1})] - \mathcal{L}(x_t) = -\kappa\eta\langle\nabla\mathcal{L}(x_t), \frac{1}{C}R^\top R\sum_{c=1}^C \mathbb{E}[\tilde{\Delta}_t^c]\rangle + \frac{\kappa^2\eta^2}{2}\mathbb{E}[(\frac{1}{C}R^\top R\sum_{c=1}^C \tilde{\Delta}_t^c)^\top \hat{H}_\mathcal{L}(\frac{1}{C}R^\top R\sum_{c=1}^C \tilde{\Delta}_t^c)]$$

$$= -\kappa\eta\langle\nabla\mathcal{L}(x_t), \frac{1}{C}\sum_{c=1}^C R^\top R\mathbb{E}[\tilde{\Delta}_t^c] - \mathbb{E}[\tilde{\Delta}_t^c]\rangle - \kappa\eta\langle\nabla\mathcal{L}(x_t), \frac{1}{C}\sum_{c=1}^C \mathbb{E}[\tilde{\Delta}_t^c]\rangle$$

$$+ \frac{\kappa^2\eta^2}{2}\mathbb{E}[(\frac{1}{C}R^\top R\sum_{c=1}^C \tilde{\Delta}_t^c)^\top \hat{H}_\mathcal{L}(\frac{1}{C}R^\top R\sum_{c=1}^C \tilde{\Delta}_t^c)]$$

$$\leq \frac{\kappa\eta K}{2}\frac{\log^{1.5}(d/\delta)}{\sqrt{b}}\|\nabla\mathcal{L}\|^2 + \frac{\kappa\eta}{2K}\frac{\log^{1.5}(d/\delta)}{\sqrt{b}}\|\frac{1}{C}\sum_{c=1}^C \mathbb{E}[\tilde{\Delta}_t^c]\|^2 - \frac{\kappa\eta K}{2}\|\nabla\mathcal{L}\|^2 - \frac{\kappa\eta}{2K}\|\frac{1}{C}\sum_{c=1}^C \mathbb{E}[\tilde{\Delta}_t^c]\|^2$$

$$+ \frac{\kappa\eta K}{2}\|\nabla\mathcal{L} - \frac{1}{K}\frac{1}{C}\sum_{c=1}^C \mathbb{E}[\tilde{\Delta}_t^c]\|^2 + \frac{\kappa^2\eta^2}{2}\mathbb{E}[(\frac{1}{C}R^\top R\sum_{c=1}^C \tilde{\Delta}_t^c)^\top \hat{H}_\mathcal{L}(\frac{1}{C}R^\top R\sum_{c=1}^C \tilde{\Delta}_t^c)]$$

$$\leq -(1 - \frac{\log^{1.5}(d/\delta)}{\sqrt{b}})\frac{\kappa\eta K}{2}\|\nabla\mathcal{L}\|^2 + \frac{\kappa\eta K}{2}\|\nabla\mathcal{L} - \frac{1}{K}\frac{1}{C}\sum_{c=1}^C \mathbb{E}[\tilde{\Delta}_t^c]\|^2$$

$$+ \frac{\kappa^2\eta^2}{2}\mathbb{E}[(\frac{1}{C}R^\top R\sum_{c=1}^C \tilde{\Delta}_t^c)^\top \hat{H}_\mathcal{L}(\frac{1}{C}R^\top R\sum_{c=1}^C \tilde{\Delta}_t^c)]$$

$$\leq -\frac{\kappa\eta K}{4}\|\nabla\mathcal{L}\|^2 + \frac{\kappa\eta K}{2}\|\nabla\mathcal{L} - \frac{1}{K}\frac{1}{C}\sum_{c=1}^C \mathbb{E}[\tilde{\Delta}_t^c]\|^2$$

$$+ \frac{\kappa^2\eta^2}{2}\mathbb{E}[(\frac{1}{C}R^\top R\sum_{i=1}^C \tilde{\Delta}_{t,i}^c)^\top \hat{H}_\mathcal{L}(\frac{1}{C}R^\top R\sum_{i=1}^C \tilde{\Delta}_{t,i}^c)], \ w.p. \ 1 - \Theta(\delta)$$

where the first inequality is directly from Lemma B.1. The second and last inequalities are from the condition of $b \geq 4\log^3(d/\delta)$. $\qquad\square$

### D.2 PROOF OF THEOREM 3.2

The first order term in Lemma 3.1 can be handled by

$$\|\nabla\mathcal{L} - \frac{1}{K}\frac{1}{C}\sum_{c=1}^C \mathbb{E}[\tilde{\Delta}_t^c]\| \leq \|\nabla\mathcal{L} - \frac{1}{K}\frac{1}{C}\sum_{c=1}^C \mathbb{E}[\Delta_t^c]\| + \frac{1}{K}\|\frac{1}{C}\sum_{c=1}^C \mathbb{E}[\Delta_t^c] - \frac{1}{C}\sum_{c=1}^C \mathbb{E}[\tilde{\Delta}_t^c]\|$$

$$\leq \frac{\eta L}{KC}\sum_{c=1}^C\sum_{i=1}^K \mathbb{E}[\|\nabla\mathcal{L}_{t,k}^c\|] + \frac{1}{KC}\sum_{c=1}^C \mathbb{E}[\|\Delta_t^c\|1_{\{\frac{1}{C}\sum_{c=1}^C\|\Delta_t^c\|\geq\tau\}}]$$

$$\leq \eta KLG + K^{\alpha-1}G^\alpha\tau^{1-\alpha},$$

where the last inequality follows by

$$\frac{1}{C}\sum_{c=1}^C \mathbb{E}[\|\Delta_t^c\|1_{\{\frac{1}{C}\sum_{c=1}^C\|\Delta_t^c\|\geq\tau\}}] = \mathbb{E}[\frac{1}{C}\sum_{c=1}^C \|\Delta_t^c\|1_{\{\frac{1}{C}\sum_{c=1}^C\|\Delta_t^c\|\geq\tau\}}]$$

$$= \mathbb{E}[(\frac{1}{C}\sum_{c=1}^C \|\Delta_t^c\|)^\alpha(\frac{1}{C}\sum_{c=1}^C \|\Delta_t^c\|)^{1-\alpha}1_{\{\frac{1}{C}\sum_{c=1}^C\|\Delta_t^c\|\geq\tau\}}] \leq (KG)^\alpha\tau^{1-\alpha}.$$

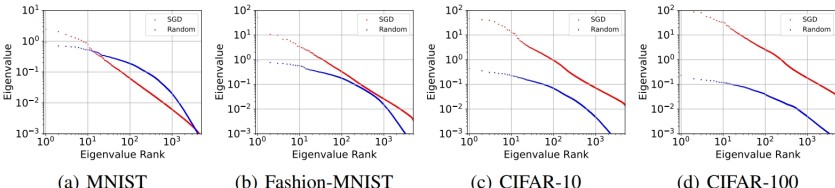

(a) MNIST (b) Fashion-MNIST (c) CIFAR-10 (d) CIFAR-100

Figure 5: The power-law structure of the Hessian spectrum on LeNet. Quoted from Fig.1 (Xie et al., 2022).

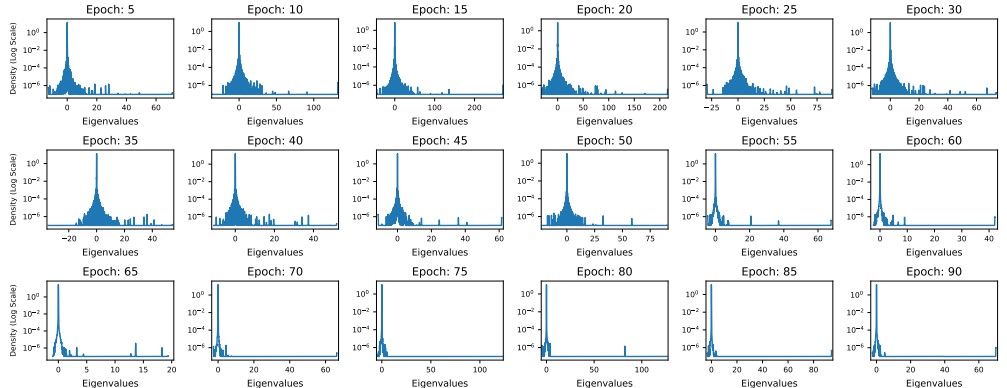

Figure 6: Eigenspectrum density every 5 epochs. The model is ViT-Small and trained on CIFAR10. The majority of eigenvalues concentrates near 0 and the density enjoys a super fast decay with the absolute values of eigenvalues, indicating a summable eigenspectra.

The second order term can be handled as follows. With probability $1 - \Theta(\delta)$,

$$\mathbb{E}[(\frac{1}{C}R^\top R \sum_{i=1}^{C} \tilde{\Delta}_{t,i}^c)^\top \hat{H}_{\mathcal{L}}(\frac{1}{C}R^\top R \sum_{i=1}^{C} \tilde{\Delta}_{t,i}^c)] = \mathbb{E}[\sum_{j=1}^{d} \lambda_j \langle \frac{1}{C}R^\top R \sum_{i=1}^{C} \tilde{\Delta}_{t,i}^c, v_j \rangle^2]$$

$$\leq \mathbb{E}[\sum_{j=1}^{d} \lambda_j 1_{\lambda_j \geq 0} \langle \frac{1}{C}R^\top R \sum_{i=1}^{C} \tilde{\Delta}_{t,i}^c, v_j \rangle^{2-\alpha} \langle \frac{1}{C}R^\top R \sum_{i=1}^{C} \tilde{\Delta}_{t,i}^c, v_j \rangle^{\alpha}]$$

$$\leq \mathbb{E}\left[\sum_{j=1}^{d} \lambda_j 1_{\lambda_j \geq 0} \left((1 + \frac{\log^{1.5}(d/\delta)}{\sqrt{b}})\frac{\tau}{\frac{1}{C}\sum_{c=1}^{C}\|\Delta_t^c\|}\frac{1}{C}\sum_{c=1}^{C}\|\Delta_t^c\|\right)^{2-\alpha}\left(\frac{1}{C}(1 + \frac{\log^{1.5}(d/\delta)}{\sqrt{b}})\sum_{i=1}^{C}\|\Delta_t^c\|\right)^{\alpha}\right]$$

$$\leq (1 + \frac{\log^{1.5}(d/\delta)}{\sqrt{b}})\hat{L}K^2\tau^{2-\alpha}G^{\alpha},$$

where the first equation follows by using the eigen-decomposition of $\hat{H}_{\mathcal{L}}$ and the second order term can be reduced to a squared inner product term. The primary trick thereafter (in the first inequality) is to split the inner product terms into two parts, which can be handled by the two-sided adaptive learning rates respectively. By applying the bounded second moment of random sketching, we find the first part with order $2 - \alpha$ is contained in a $(1 + \frac{\log^{1.5}(d/\delta)}{\sqrt{b}})\tau-$ball with high probability, and the second part with order $\alpha$ is bounded by applying Assumption 5. Then Theorem 3.2 follows by combining the first order term and the second-order term by union bounds, as well as applying the condition of $b \geq 4\log^3(d/\delta)$.

## E  EXPERIMENTAL DETAILS AND ADDITIONAL RESULTS

Aside from the experimental configurations described in the main paper, we provide additional details.

For the sketched adaptive FL methods. The server optimizer. We use Cross Entropy with label smoothing as the loss function. The parameter for label smoothing is 0.1. We use a cosine learning rate scheduler on the server optimizer, with the minimal learning rate is $1e - 5$. Client batch size is

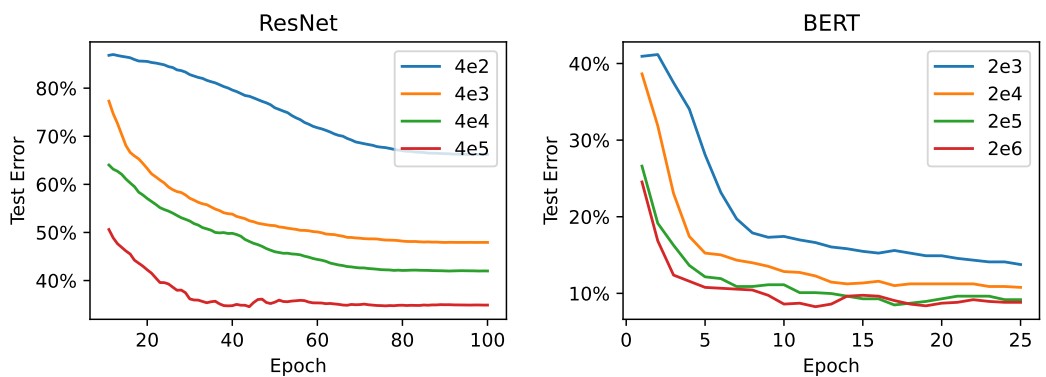

Figure 7: Comparing the performance of tiny sketch sizes on ResNet and BERT. The experiment settings are the same as in Fig. 1 and Fig. 3
.

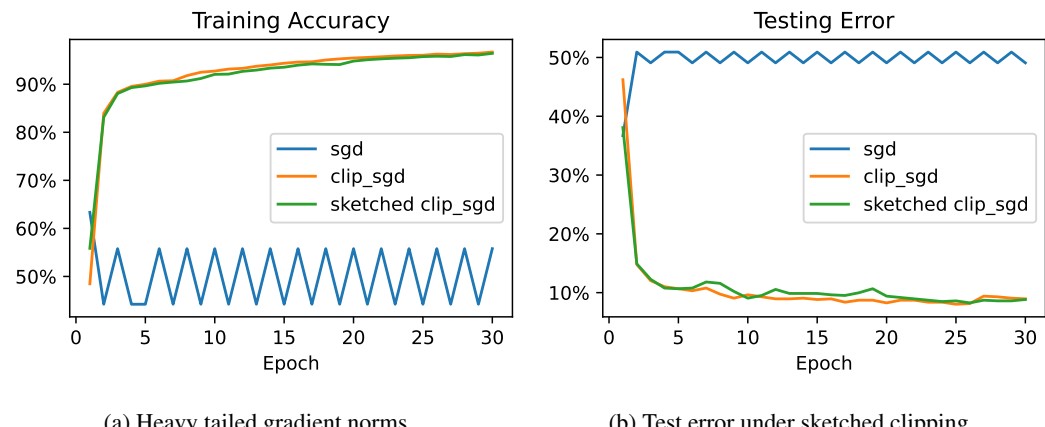

(a) Heavy tailed gradient norms.

(b) Test error under sketched clipping.

Figure 8: Sketched Clipping Methods on BERT: (a) Training Accuracy; (b) Testing Error. Under the same hyperparameters, plain SGD does not converge, while clip SGD and its sketched variant converge and generalize. Sketched Clip SGD achieves comparable performance as the unsketched Clip SGD.

128, and weight decay is $1e-4$. For SGD and SGDm methods, the learning rate is 1.0. For SGDm, the momentum is 0.9. For Adam optimizer, the learning rate is 0.01, and the momentum is 0.9. The learning rates are tuned to achieve the best performance. We adopt SRHT as the sketching operator. The SRHT matrix $R$ can expressed as $R = \sqrt{n}/bSHD$, where $S \in \mathbb{R}^{b \times n}$ is a random matrix whose rows are $b$ uniform samples (without replacement) from the standard basis of $R^n$. $H \in \mathbb{R}^{n \times n}$ is a normalized Walsh-Hadamard matrix, and $D \in \mathbb{R}^{n \times n}$ is a diagonal matrix whose diagonal elements are i.i.d. Rademacher random variables.

Our experiments were conducted on a computing cluster with AMD EPYC 7713 64-Core Processor and NVIDIA A100 Tensor Core GPU.

To verify Assumption 4, we plot the full Hessian eigenspectrum throughout the training process in Fig. 6. We used stochastic lanczos algorithm implemented by the pyHessian library (Yao et al., 2020) to approximate the distribution of the full eigenspectrum. Our main claim in Assumption 4 is that the Hessian eigenspectrum at an iterate is summable and the sum is independent of the ambient dimension, which can be satisfied by common distributions, like power-laws. We run testing experiments on ViT-small and train on CIFAR-10 dataset, with sketched Adam optimizer. In Fig. 6, we see the majority of eigenvalues concentrates near 0. The density enjoys a super fast decay with the absolute values of eigenvalues. The decay also holds throughout the training process. This empirical evidence shows the validity of our assumption.

In the main body of the paper, we have achieved 99.9% compression rate and 99.98% compression rate for ResNet and BERT respectively. We further include the results on smaller $b$ in Fig. 7. In principle, an extremely tiny sketch size (with 400 in vision tasks and 2000 in language tasks) still converges but generates an unfavorable local minima that hardly generalizes.

In the following, we present another empirical result on the clipping method. The goal here is to show the superiority of (sketched) clipping methods over the plain SGD optimizer. We run BERT model on SST2 dataset. The dataset is split among 5 distinct clients in an i.i.d way. The normalization factor in the clipping method is set as 0.03. In Figure. 8, we show that (sketched) clip SGD method has better performance in convergence and generalization, while the plain SGD method fails to converge. It is also observed that sketching does not cause drop in testing error.

# F  ADDITIONAL DISCUSSION ON EXPERIMENTS

We added two recent approaches, CD-Adam (Wang et al., 2022) and CocktailSGD (Wang et al., 2023), which are representative of state-of-the-art adaptive methods and SGD-based methods representatively. In Table 1 we compare the performance of baseline methods and sketched Adam, and derive two takeaways:

- On the vision task (CIFAR-10), sketched Adam significantly outperforms both CD-Adam and CocktailSGD.

- On the language task (SST2), sketched Adam are close with CocktailSGD, which is originally designed for training LLMs. Other algorithms fall short.

We select the learning rate under strict hyperparameter tuning protocols. We split the dataset into train/val/test sets with 10:1:1 on CIFAR-10 and 40:1:2 on SST2 (the default split). We tune the hyperparameters based on the performance over the validation set. For CocktailSGD, we adopt the default compression setting, i.e. 20% random sparsification, 10% top-k compression and 4-bit quantization, which amounts to approximately 99% compression rate. We make sure the optimal learning rate is strictly within the test interval, i.e. not on the boundary. The error rate curves on the validation set are shown in Fig. 9 and Fig. 10.

We also conduct new experiments to assist the exposition of heavy-tailed noise. In Figure 11, we plot the stochastic gradient. More specifically, we fix the client model (at the end of each local training step) and iterate over the local minibatches to collect the stochastic gradient norm. We fit the distribution with a Levy distribution in each subplot. The plot indicates it's not rare to encounter heavy-tailed noise in client model and hence consolidates Assumption 5.

|  | Adam | FetchSGD | CocktailSGD | CD-Adam | 1Bit-Adam |
|---|---|---|---|---|---|
| CIFAR-10 | 22.6% | 28.4% | 25.9% | 25.7% | 25.5% |
| SST2 | 7.80% | 10.6% | 8.03% | 8.83% | 9.06% |

Table 1: Error Rate on Test Set.

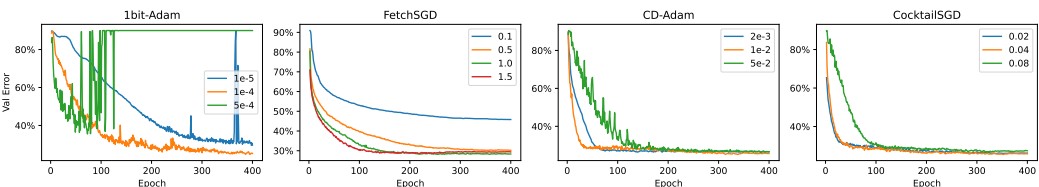

Figure 9: Validation Error on CIFAR-10. The setting is same as Fig. 2. Sketch Size $b = 4e5$.

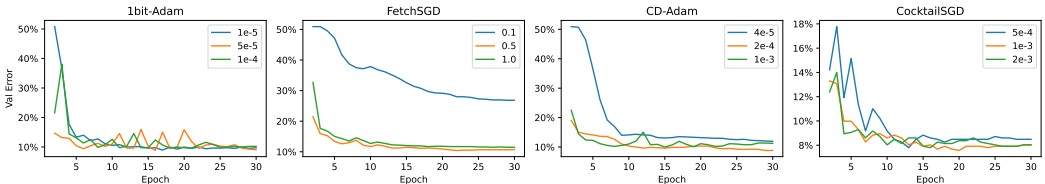

Figure 10: Validation Error on SST2. The setting is same as Fig. 3. Sketch Size $b = 2e6$.

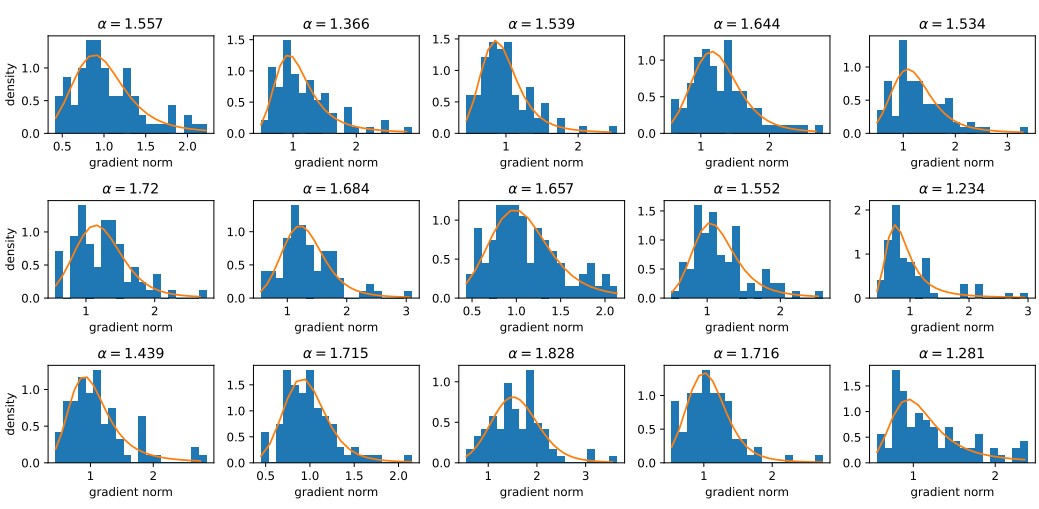

Figure 11: Histogram of stochastic gradient norm evaluated on identical local model parameter. The orange curve is the pdf of Levy distribution whose $\alpha$ is in the title. Each subplot represents one local model.

## G  Convergence Without Bounded Gradient Norm Assumption

In this section, we prove the convergence of SAFL (Algorithm. 5) under a simplified scheme. To better focus on the gradient norm, we adopt gradient descent (deterministic) updates on each client. We also restrict the local step $K$ to be 1. The proof follows the general idea recently proposed in (Li et al., 2024).

First, we derive the gradient norm bound affine to the loss function based on smoothness in Lemma G.1.

**Lemma G.1.** *For any $L$-smooth function $\mathcal{L}(x)$, $\|\nabla\mathcal{L}(x)\|^2 \leq 2L(\mathcal{L}(x) - \mathcal{L}^*)$.*

*Proof.* Let $y = x - \frac{1}{L}\nabla\mathcal{L}(x)$. Then we have

$$\mathcal{L}^* - \mathcal{L}(x) \leq \mathcal{L}(y) - \mathcal{L}(x) \leq \langle\nabla f(x), y - x\rangle + \frac{L}{2}\|y - x\|^2 = -\frac{1}{2L}\|\nabla\mathcal{L}(x)\|^2.$$

Rearranging the terms yields the lemma. $\qquad\square$

We rewrite the descent lemma under the specific condition, which is a direct derivation from Lemma 2.2.

**Lemma G.2.** *For any round $t \in [T]$, there exists function $\Phi_t \geq 0$ such that*

$$\mathcal{L}(z_{t+1}) \leq \mathcal{L}(z_t) + \Phi_t - \Phi_{t+1} - \frac{\kappa\eta}{C}\sum_{c=1}^{C}\nabla\mathcal{L}(x_t)^\top\hat{V}_{t-1}^{-1/2}R_t^\top R_t\nabla\mathcal{L}^c(x_t) + (z_t - x_t)^\top H_\mathcal{L}(\hat{z}_t)(z_{t+1} - z_t),$$

*where $H_\mathcal{L}(\hat{z}_t)$ is the loss Hessian at some $\hat{z}_t$ within the element-wise interval of $[x_t, z_t]$.*

Let $G$ be a constant which will be specified later. Let $F = \frac{1}{2L}G^2$. Denote the optimization horizon (server steps) by $T$. Denote $\hat{t} = \min\{t|\mathcal{L}(z_t) - \mathcal{L}^* > F\} \wedge (T + 1)$. We consider the case when $\hat{t} \leq T$. For $t < \hat{t}$, we have $\mathcal{L}(x_t) - \mathcal{L}^* \leq F$ and thus $\|\nabla\mathcal{L}(x_t)\| \leq G$, which guarantees an upper bound on the gradient in the restricted region. We follow the technique in the bounded gradient setting. For any $t \leq \hat{t}$, with probability $1 - 2\delta$, the first order term can be lower bounded by

$$\nabla\mathcal{L}(x_t)^\top\hat{V}_{t-1}^{-1/2}R_t^\top R_t\nabla\mathcal{L}(x_t)$$

$$\geq (1 - \frac{\log^{1.5}(Cd/\delta)}{\sqrt{b}})\|\nabla\mathcal{L}(x_t)\|\|\hat{V}_{t-1}^{-1/2}\nabla\mathcal{L}(x_t)\|$$

$$\geq (1 - \frac{\log^{1.5}(Cd/\delta)}{\sqrt{b}})\left(\sqrt{1 + \frac{\log^{1.5}(Ctd^2/\delta)}{\sqrt{b}}}\eta G + \epsilon\right)^{-1}\|\nabla\mathcal{L}(x_t)\|^2,$$

where the second inequality follows by Lemma. 2.4.

The second order term can be bounded by

$$(z_t - x_t)^\top H_{\mathcal{L}}(\hat{z}_t)(z_{t+1} - z_t)$$

$$=(\frac{1}{1-\beta_1}(x_{t+1} - x_t) - \frac{\beta_1}{1-\beta_1}(x_t - x_{t-1}))^\top \hat{H}_{\mathcal{L}} \frac{\beta_1}{1-\beta_1}(x_t - x_{t-1})$$

$$\leq(1 + \frac{\log^{1.5}(Cd/\delta)}{\sqrt{b}})^2 \eta^2/\varepsilon^2 \frac{\beta_1 + \beta_1^2}{(1-\beta_1)^2} \sum_{i=1}^d |\lambda_i|(\sum_{\tau=0}^t (1-\beta_1)\beta_1^{t-\tau}\|g_\tau\|)(\sum_{\tau=0}^{t-1}(1-\beta_1)\beta_1^{t-1-\tau}\|g_\tau\|)$$

$$=(1 + \frac{\log^{1.5}(Cd/\delta)}{\sqrt{b}})^2 \eta^2/\varepsilon^2 \beta_1(\beta_1+1)\hat{L} \sum_{\tau_1=0}^t \sum_{\tau_2=0}^{t-1} \beta_1^{2t-1-\tau_1-\tau_2}\|g_{\tau_1}\|\|g_{\tau_2}\|$$

$$\leq(1 + \frac{\log^{1.5}(Cd/\delta)}{\sqrt{b}})^2 \eta^2/\varepsilon^2 \frac{\beta_1(\beta_1+1)}{2}\hat{L} \sum_{\tau_1=0}^t \sum_{\tau_2=0}^{t-1} \beta_1^{2t-1-\tau_1-\tau_2}(\|g_{\tau_1}\|^2 + \|g_{\tau_2}\|^2)$$

$$=(1 + \frac{\log^{1.5}(Cd/\delta)}{\sqrt{b}})^2 \eta^2/\varepsilon^2 \frac{\beta_1(\beta_1+1)}{2}\hat{L}(\sum_{\tau_1=0}^t \beta_1^{t-\tau_1}\|g_{\tau_1}\|^2(\sum_{\tau_2=0}^{t-1}\beta_1^{t-1-\tau_2}) + \sum_{\tau_2=0}^{t-1}\beta_1^{t-1-\tau_2}\|g_{\tau_2}\|^2(\sum_{\tau_1=0}^t \beta_1^{t-\tau_1}))$$

$$\leq(1 + \frac{\log^{1.5}(Cd/\delta)}{\sqrt{b}})^2 \eta^2/\varepsilon^2 \frac{\beta_1(\beta_1+1)}{2(1-\beta_1)}\hat{L}(\sum_{\tau_1=0}^t \beta_1^{t-\tau_1}\frac{1}{1-\beta_1}\|g_{\tau_1}\|^2 + \sum_{\tau_2=0}^{t-1}\beta_1^{t-1-\tau_2}\frac{1}{1-\beta_1}\|g_{\tau_2}\|^2)$$

Plugging the first-order term and second-order term back to the descent lemma, and apply $b = \frac{1}{b_0^2}\log^3(CTd^2/\delta)$, where $b_0$ is arbitrary constant smaller than 1. We have

$$\mathcal{L}(z_{t+1}) + \Phi_{t+1} \leq \mathcal{L}(z_t) + \Phi_t - \kappa\eta(1-b_0)(\sqrt{1+b_0}\eta G + \epsilon)^{-1}\|\nabla\mathcal{L}(x_t)\|^2$$

$$+ \kappa^2\eta^2(1+b_0)^2 \frac{\beta_1(\beta_1+1)\hat{L}}{2(1-\beta_1)\epsilon^2}(\sum_{\tau_1=0}^t \beta_1^{t-\tau_1}\|g_{\tau_1}\|^2 + \sum_{\tau_2=0}^{t-1}\beta_1^{t-1-\tau_2}\|g_{\tau_2}\|^2)$$

Summing the descent inequalities up across different iterations yields

$$\mathcal{L}(z_{t+1}) + \Phi_{t+1}$$

$$\leq\mathcal{L}(z_0) + \Phi_0 - \sum_{\tau=0}^t \eta\kappa(1-b_0)(\sqrt{1+b_0}\eta G + \epsilon)^{-1}\|\nabla\mathcal{L}(x_\tau)\|^2$$

$$+ \kappa^2\eta^2(1+b_0)^2\frac{\beta_1\hat{L}}{2(1-\beta_1)\epsilon^2} \sum_{\tau=0}^t (\sum_{\tau_1=0}^{t-\tau}\beta_1^{t-\tau-\tau_1})\|\nabla\mathcal{L}(x_\tau)\|^2$$

$$\leq\mathcal{L}(z_0) + \Phi_0 - \sum_{\tau=0}^t \eta\kappa(1-b_0)(\sqrt{1+b_0}\eta G + \epsilon)^{-1}\|\nabla\mathcal{L}(x_\tau)\|^2 + \kappa^2\eta^2(1+b_0)^2\frac{\beta_1(1+\beta_1)\hat{L}}{2(1-\beta_1)^2\epsilon^2} \sum_{\tau=0}^t \|\nabla\mathcal{L}(x_\tau)\|^2$$

$$\tag{3}$$

Let $t+1 = \hat{t}$. We have $\mathcal{L}(z_{t+1}) - \mathcal{L}^* > F := \frac{1}{2L}G^2$ by definition. On the other hand, by the descent lemma, with sufficiently small $\kappa$, we can guarantee

$$\mathcal{L}(z_{t+1}) - \mathcal{L}^* + \Phi_{t+1} \leq \mathcal{L}(z_0) - \mathcal{L}^* + \Phi_0 := \Delta_0$$

where $\Delta_0$ is bounded given the initialization is benign. We specify $G$ as any constant larger than $2L\Delta_0$ which will yield contradiction. Hence we conclude that along the optimization trajectory, the norm of gradient is upper bounded by $G$. By Eq. 3, we can also derive the convergence result on the relaxed assumption.

**Theorem G.3.** *Suppose the sequence of iterates $\{x_t\}_{t=1}^T$ is generated by Algorithm 1 (SAFL) with a constant learning rate $\eta$ and $\kappa$ subject to $\kappa < (1-\beta_1)^2\epsilon^2\eta(1-b_0)((1+b_0)^2\beta_1(1+\beta_1)\hat{L}(\sqrt{1+b_0}\eta G + \epsilon))^{-1}$. Set $G = 2L\Delta_0 + 1$. Set $b = \frac{1}{b_0^2}\log^3(CTd^2/\delta)$, where $b_0 \in (0,1)$ is an arbitrary constant. Under Assumptions 4, for any $T$ and $\epsilon > 0$, with probability $1 - \Theta(\delta)$,*

$$\frac{1-b_0}{2(\sqrt{1+b_0}\eta G + \epsilon)} \sum_{t=0}^T \|\nabla\mathcal{L}(x_t)\|^2 \leq \mathcal{L}(z_0) + \Phi_0.$$

| Algorithms | Communication Bits | learning rate | Convergence Rate |
|---|---|---|---|
| FetchSGD | $\tilde{O}(1)$ | $O(1/\sqrt{T})$ | $O(1/\sqrt{T})$ [A] |
| CocktailSGD | $O(1)$ | $O(1/(\sqrt{T}+T^{1/3}d^2+d^3))$ | $O(1/\sqrt{T}+d^2/(T)^{2/3})$ |
| CD-Adam | $O(1)$ | $O(1/\sqrt{d})$ | $O(\sqrt{d}/\sqrt{T})$ |
| Onebit-Adam | $O(d)$ | $O(1/\sqrt{T})$ | $O(1/\sqrt{T})$ |
| Efficient-Adam | $O(1)$ | $O(1/\sqrt{T})$ | $O(\sqrt{d}/\sqrt{T})$ [B] |
| Ours | $\tilde{O}(1)$ | $O(1/\sqrt{T})$ | $O(1/\sqrt{T})$ [C] |

Table 2: Comparison on Theoretical Guarantees. We only include the dependence on $d$ and $T$. (A) Needs a heavy-hitter assumption, otherwise deteriorated to $O(T^{1/3})$. (B) There is no asymptotic convergence for the algorithm. (C) requires the assumption on the fast-decay Hessian eigenspectrum. Otherwise, the convergence rate can deteriorate to $O(d/\sqrt{T})$ under dimension-independent learning rate.

The two simplifications applied to the analysis does not harm the generalizability of the theorem. First, if the client performs multi-step gradient descent in the local training phase, we additionally need a guarantee on the norm of all subsequent local gradients. Notice that the analysis in (Li et al., 2024) are not specific to any architectures or data distribution, we can use the same technique to show boundedness of the gradient norm along the optimization trajectory over each client. Second, the case involving stochastic noise has been considered in (Li et al., 2024). Instead of showing the deterministic decrease in $\mathcal{L}(z_t) + \Phi_t$, it is advocated to alternatively show a high probability decrease given the stochastic noise is bounded with high probability, which is exactly what we managed to show in the main paper. Therefore, this improved technique can be seamlessly applied to our analysis.

# H    DISCUSSION ON ASSUMPTIOM 4 AND RELATED WORKS

Assumption 4 is one of the key assumptions in our theoretical improvement. However, the anisotropic structure in Hessian mainly helps in dealing with the second order term. Solely applying Assumption 4 is not sufficient. First, the usage of assumption is highly specific to the compression operator in this paper, i.e. random sketching. Previous works fail to utilize the anisotropic Hessian structure in deep learning. For example, in (Wang et al., 2023), Lemma A.1 indicates the discrepancy between the local and the global model unavoidably picks up a dimension dependence because of the accumulation of the error introduced by their specific compression algorithm. This accumulative effect cannot be handled by simply applying the Hessian assumption. In (Wang et al., 2022), the dimensional dependence arises in their first-order descent term (B.14) and (B.15), and hence the assumption on Hessian does not apply either.

Additionally, we summarize the theoretical guarantees of the existing approaches in Table 2. From the table, we can see all the comparisons made in the main paper are fair.

