# OpenReview forum: "Sketched Adaptive Federated Deep Learning: A Sharp Convergence Analysis"
_ICLR.cc/2025/Conference — Submitted to ICLR 2025_

### Official Review · Reviewer_D3d5 · 2024-10-22

**Soundness:** 3
**Presentation:** 2
**Contribution:** 3
**Rating:** 6
**Confidence:** 4

**Summary:**

This paper considers sketching-based federated learning algorithms, the main contributions are theoretical improvements over [Rothchild et al., 2020; Song et al., 2023] and empirically validations of their theory. It proposes an FL algorithm that utilizes sketching to compress the communication while enabling advanced optimizers beyond SGD. From a convergence rate perspective, it shows that under stronger assumptions, it is possible to achieve ${\rm poly}\log d$ dependence in the iteration complexity instead of $d$ in previous works. It also performs a thorough study on the performance of mixing sketching with stochastic gradient plus noisy second order information. Empirically, it shows superior performance over [Rothchild et al., 2020] on CIFAR10 and GLUE.

**Strengths:**

This paper greatly extends the theoretical analysis of [Rothchild et al., 2020; Song et al., 2023] that uses sketching to improve the communication cost. Specifically, it gives a very general analysis in which adaptive optimizers are allowed and it could incorporate elementwise second moment of the sketched gradient. The analysis is involved as there are 3 types of errors: sketching error, local steps taken by client training and client data distribution. In particular, allowing client to do adaptive optimization coupled with SGD faces many complications caused by sketching. This paper gives a good error decomposition and uses a Martingale to handle the challenge. It further considers the setting where the bounded gradient norm (square) is relaxed to bounded $\alpha$-moment for $\alpha\in (1, 2]$. It is important to improve the dimension dependence for the convergence in the previous work, as what is observed in practice is that the sketching dimension has a far less impact on the overall convergence than causing a linear in dimension blowup. Finally, experiments are performed to show the effectiveness of the theory.

**Weaknesses:**

It is worth noting that removal of linear dependence on $d$ is a consequence of stronger assumption rather than an advancement of analysis, as in previous works, they only assume basic smoothness and bounded gradient (note that [Song et al., 2023] didn't assume bounded gradient) which only says that the spectral norm of the Hessian is at most $L$, while in this work the Hessian matrix eigenspectrum assumption in addition requires the nuclear norm of the Hessian is at most some $\hat L$. Note that in the worst case, $\hat L=d L$, which I believe is the result of [Song et al., 2023]. Arguably, in many scenarios it is true that the Hessian only has a few large eigenvalues, but I think it's important to point out the impact of the assumption, and compare \& contrast with previous works about them. I would encourage the authors to further clarify and compare the results about assumptions and convergence in a table with previous works (e.g., for comparing with [Rothchild et al., 2020], one could use gradient instead of stochastic gradient, take the optimizer to be the identity mapping, then does the convergence improves upon it due to nuclear norm assumption? For comparing with [Song et al., 2023], one might consider further remove the bounded gradient assumption and see whether the analysis still goes through).

**Questions:**

See weaknesses. Also, how does bounded gradient norm or bounded $\alpha$-moment play a role in the analysis? Is it possible to remove them while retaining nuclear norm bound assumption, and achieve a result similar to [Song et al., 2023] by replacing the $dL$ factor with $\hat L$?

---

### Official Review · Reviewer_LLK4 · 2024-10-28

**Soundness:** 3
**Presentation:** 4
**Contribution:** 1
**Rating:** 3
**Confidence:** 4

**Summary:**

This paper introduces Sketched Adaptive Federated Learning (SAFL), which combines gradient sketching methods in FL with adaptive optimizers (e.g. Adam). It is proven that the proposed framework can converge with (1) sub-gaussian noise and (2) heavy-tailed noise (with a clipped variant), and the dependence on the dimension $d$ is only logarithmic. Experiments show that Sketched Adam is on par or better than other sketched baselines for training ResNet, ViT, and BERT.

**Strengths:**

1. The paper is well-written. The theoretical motivation and contributions of this work are all clearly stated.
2. The high-level problem of reducing communication cost in federated learning is important, especially when training deep networks.

**Weaknesses:**

1. The improved theoretical results compared to prior work may be a direct result of additional assumptions.

    1a. The biggest assumption here compared to prior work is the decay of the Hessian eigenspectrum (Assumption 4), and it seems that the reduced dependence on $d$ (from polynomial to logarithmic) may come as an immediate consequence of this assumption (Lines 299-313). The reduced $d$ dependence is the main theoretical improvement claimed by the authors, but I'm not sure whether there are any real technical challenges to achieve this reduced dependence, other than applying Assumption 4.

    1b. It is unclear whether the constant $\hat{L}$ changes with $d$ in practice, which seems very possible to me. If so, then there is really some hidden dependence on $d$ in the constant $\hat{L}$.

    1c. The combination of assumptions is not well motivated, neither theoretically nor empirically. Specifically, the paper assumes bounded gradients (Assumption 1) together with Hessian eigenspectrum decay (Assumption 4). Deep neural networks may satisfy Assumption 4, but Assumption 1 definitely does not hold for such networks. On the other hand, some classic optimization problems in ML satisfy Assumption 1 (e.g. logistic regression), but such examples fall outside of deep learning, which is the motivating example for Assumption 4. I don't see any reason to consider these assumptions together other than ease of analysis.

2. The experiments don't contain significant contributions.

    2a. Sketched Adam does outperform other sketched optimizers (e.g. sketched sgdm), but I don't interpret this as a contribution of this paper, since this is essentially a comparison of Adam against SGDM and SGD. The improvement against prior work (1-bit Adam, FetchSGD), is a minor contribution to me, since the novelty of Sketched Adam is somewhat low.

    2b. The experiments fail to compare against any prior work from the last three years. 1-bit Adam and FetchSGD are both old algorithms at this point. The other parts of the paper cite Efficient-Adam (Chen et al, 2022), Compressed AMSGrad (Wang et al, 2022), CocktailSGD (Wang et al, 2023), and many others, but recent baselines are missing from the experimental comparison.

    2c. The methodology around evaluation is somewhat vague. Line 1351 says "The learning rates are tuned to achieve the best performance", and this is all of the information about hyperparameter tuning. It seems possible that there was no strict hyperparameter protocol, which invites the possibility that some algorithms were tuned more than others. Also, there is no mention of multiple random seeds, so it seems that all experiments include results over a single random seed.

3. Weak motivation for heavy-tailed noise in federated learning. A constant theme throughout the paper is the importance of heavy-tailed noise in federated learning, as observed in (Yang et al, 2022). However, I am skeptical about this connection. (Yang et al, 2022) observes a heavy-tailed distribution of \textit{local client updates aggregated across an entire round}, but they use this as justification to assume a heavy tailed distribution of stochastic gradient noise \textit{at a single point}. I don't agree with this justification at all, which appears to confuse the distribution of local client updates aggregated across an entire round with the distributino of stochastic gradient noise at each individual point. The current submission uses this observation as inspiration for the connection between FL with heterogeneous data and heavy-tailed noise, e.g. on Line 048, where it is claimed that adaptive methods are better for FL with heterogeneous data due to heavy-tailed noise. These claims are not justified, in my opinion, and this severely undermines the motivation for a large part of this submission.

4. Unclear/missing comparisons with previous work.

    4a. The paper contains very little comparison of the current theoretical work against prior work. The only such comparison is the claim that previous works have a linear dependence on $d$, compared to the logarithmic dependence on $d$ achieved by this paper. However, the convergence rates achieved by the previous papers are not explicitly written here, and the explanation provided for the $d$ dependence of prior work (Lines 074-076) relies on the condition of constant per-round communication cost as $d$ scales. It is unclear whether this is an apples-to-apples comparison against the convergence rate of the current work. I recommend that the authors include a more clear, direct comparison of their convergence rate against that of prior works.

    4b. The experiments are missing many baselines (see Weakness #2).

    4c. The discussion of related work on heavy tailed noise (Lines 355-365) should be amended. First, I did not see any mention of heavy tailed noise in (Charles et al, 2021), so I recommend a more specific discussion of how this work related to yours. Also, I recommend clarifying the sentence at the end of this paragraph, specifically the claim of optimality. (Zhang, 2020) indeed show optimality of certain methods in the centralized case, but they don't consider the distributed case. (Liu, 2022) consider gradient clipping in the distributed case, but they do not consider heavy tailed noise or make claims of optimality.

Formatting/typos:
- The abstract should be limited to one paragraph, as written in the submission instructions.
- Line 081: "depends only on a logarithmically on the ambient dimension $d$". Should say "depends only logarithmically"?
- Line 133: "Update paramters" should say "Update parameters"
- Line 193: In property 2, I recommend modifying the definition of $\texttt{sk}$ and $\texttt{desk}$ to include $R$ and $R_t$ as parameters over which we can take expectation. Right now the expectation in property 2 is not very rigorously defined, since there are no random variables appearing in property 2 over which we could take expectation.
- Figure 1 is a little confusing to read, since the first two plots show one compression rate, the third plot shows a different compression rate, and the last plot shows results for a single algorithm with many compression rates. It would be helpful if this information were more organized. Also, I recommend using colors consistently to represent a single algorithm across different plots, as much as possible. For example, your Figure 1 uses green/blue for Adam in the first/second plot, respectively, then all colors for Adam in the last plot. Also, orange is used for sgdm in the first plot, then for 1 bit Adam in the second plot.

**Questions:**

1. What are the technical challenges of reducing the dependence of $d$ from polynomial to logarithmic, other than applying Assumption 4? Does the proof require any new techniques compared to previous work?
2. What is the motivation for simultaneously assuming bounded gradients (Assumption 1) and decaying Hessian eigenspectrum (Assumption 4)? Do you have any natural examples of ML training objectives that satisfy these two assumptions simultaneously?
3. It seems possible that $\hat{L}$ depends on $d$. Are there any experiments in the literature investigating how $\hat{L}$ changes with $d$?
4. What is the communication cost of FetchSGD compared to the other evaluated algorithms? Did you control for communication cost when comparing your algorithm to theirs?
5. Line 079: What does almost iid mean? This phrase is used a couple times without definition, then is not referenced again.
6. Line 234: "The i.i.d. data assumption leads to the bounded gradient assumption". I don't understand this claim. Many applications have i.i.d. data and unbounded gradients, for example if the global objective is quadratic.
7. What was the protocol for hyperparameter tuning? Is it possible that some algorithms were tuned more than others? How many random seeds were used to evaluate each algorithm?
8. What does Figure 4(a) show exactly? Local stochastic gradients sampled at the same point? Local stochastic gradients sampled at different points? Or local gradients sampled at different points?

---

### Official Review · Reviewer_pxMX · 2024-11-06

**Soundness:** 4
**Presentation:** 3
**Contribution:** 3
**Rating:** 6
**Confidence:** 3

**Summary:**

This paper proposes a framework for sketched adaptive federated learning (SAFL), a method for communication-efficient federated learning. In the (nearly) i.i.d. client setting, they prove a 1/sqrt(T) convergence rate for SAFL with a 1/T rate near initialization, with low (logarithmic) communication costs by using a novel assumption on the decay of the eigenspectrum of Hessians in deep neural networks. In the non i.i.d. setting, they prove convergence, even in the presence of heavy-tailed noise.

Experiments show improvements using SAFL over existing communication-efficient federated learning methods when tested on the CIFAR-10 dataset and the SST2 task.

**Strengths:**

The paper provides theoretical results on the convergence rates of the algorithm in both the i.i.d. and non-i.i.d. client settings by utilizing a novel assumption observed in other recent work. It is interesting to see strong assumptions validated in practice, and the resulting theory is of independent interest to other stochastic optimization results for deep learning. The proposed framework and analysis are highly flexible and natural and can be used with various optimizers and sketching algorithms. The framework is tested on different types of models and datasets.

The paper reads clearly and is well-motivated.

**Weaknesses:**

Assumptions 1 and 2, while standard, are strong and hard to ensure in practice. The claim that SAFL enjoys better performance near initialization due to adaptivity feels unmotivated. It is also unclear if performance improvements are primarily due to using Adam over SGD, which is used in FetchSGD. Finally, the sketching algorithm used for the experiments is unclear.

**Questions:**

1. The power law decay of the spectrum is only valid after a few epochs of SGD. Given this, is assuming that assumption 4 is true throughout the training process valid? Figure 6 shows an eigenvalue decay but does not necessarily confirm that the decay follows a power law.

2. Typically, SGD outperforms Adam in vision tasks with convnets. However, this does not seem to be the case in SAFL. Why is this occurring?

3. How feasible would it be to extend the proof to relax assumptions 1 and 2 to affine conditions, i.e. the gradient is bounded by an affine function?

4. On real data, we can see that Adam outperforms SGD when used with this framework. However, does this still occur on synthetic data when comparing AMSGrad to SGD to show that the improvement in corollary 2 is due to adaptivity? Or is there a more straightforward explanation?

5. Could the authors include the sketching algorithm used in the experiment?

6. If FetchSGD can be used with Adam, does SAFL still outperform the new “FetchAdam” algorithm?

(I am willing to raise the score if the weakness referred to above and questions are well addressed)

---

### Official Review · Reviewer_Qh7r · 2024-11-13

**Soundness:** 2
**Presentation:** 3
**Contribution:** 2
**Rating:** 3
**Confidence:** 4

**Summary:**

This paper introduces the Sketched Adaptive Federated Learning (SAFL) framework, aiming to alleviate communication burdens in federated learning (FL) by leveraging gradient sketching alongside adaptive optimizers. The authors propose that by combining techniques like CountSketch and quantization with adaptive optimizers (e.g., Adam, AMSGrad), FL can achieve improved communication efficiency. The framework is extended to handle non-i.i.d. settings via a clipped variant, SACFL, which uses gradient clipping to ensure stability even with heavy-tailed noise. The paper’s convergence results are claimed to scale logarithmically with respect to model size, which is theoretically significant for applications with large model parameters. Additionally, experiments in vision and NLP domains provide empirical validation, showing that SAFL and SACFL can maintain comparable accuracy while achieving higher compression rates.

**Strengths:**

The paper analyzes Adaptive Opt + Grad Sketching.

While adaptive optimizers (e.g., Adam) and sketching techniques have been individually studied in FL, their combined use has been relatively unexplored.

Convergence analyses showing that communication costs scale logarithmically with model dimensionality, which contrasts with prior work where communication costs are linearly dependent on model size.

This logarithmic dependency is critical for scalability and suggests that SAFL could be feasible for modern deep learning models with millions or billions of parameters.

Notably, the analysis leverages recent findings on the anisotropic Hessian spectrum in deep networks, showcasing an innovative application of this structural property in theoretical work.

The dual versions of the framework, SAFL (for i.i.d. or mild noise) and SACFL (for heavy-tailed noise), allow it to address both homogeneous and heterogeneous data environments, which is essential for federated scenarios with variable client data. This flexibility increases its potential applicability to real-world FL problems, where client data distributions often deviate from the i.i.d. assumption.
Experimental Validation Across Domains:

The paper presents a range of experiments on tasks in both vision (CIFAR-10) and NLP (SST2), validating that the framework performs well on distinct model architectures (ResNet, Vision Transformer, BERT) and data regimes. The empirical results show that SAFL maintains comparable performance to full-dimensional optimizers, which supports its practical relevance.

**Weaknesses:**

Dependency on Assumptions Regarding Hessian Eigen Spectrum Decay:

The convergence guarantees rely heavily on the assumption that the Hessian’s eigen spectrum decays sharply (anisotropy in curvature). Although this phenomenon is observed in many neural network architectures, it is not universally applicable, particularly in certain tasks, models, or data distributions. For instance, models used on structured data or non-standard architectures may not exhibit this property, limiting SAFL’s applicability in such cases. A more in-depth discussion on these limitations, or an empirical analysis of SAFL’s performance under weakly decaying Hessian spectrums, would clarify the practical applicability of the theoretical results.
Insufficient Baseline Comparisons and Ablation Studies:

The experimental comparisons are mostly limited to other communication-efficient algorithms like FetchSGD and 1bit-Adam, without sufficient benchmarking against non-adaptive FL methods (e.g., FedAvg) or ablation studies isolating the contributions of sketching and adaptivity. This makes it challenging to discern the individual impacts of gradient sketching and adaptive optimizers on communication efficiency and convergence stability. Including a wider range of baselines and detailed ablation studies would better contextualize the advantages of SAFL.

Scalability and Stability in Real-World FL Environments:

The experiments focus on setups with small client numbers and controlled conditions. Real-world FL often involves thousands of clients with highly heterogeneous data, variable network conditions, and diverse computational resources. The scalability of SAFL to large client pools, as well as its robustness under more realistic conditions, remains uncertain. A thorough examination of SAFL’s performance under these conditions, particularly with varied network latencies and client-side hardware capacities, would enhance confidence in its practical relevance.

Trade-offs Between Compression and Model Quality:

The paper briefly notes that SAFL performs well under high compression rates, but a deeper analysis of the trade-offs between compression rates and model performance is missing. High compression could potentially degrade convergence rates or accuracy, especially for tasks that are sensitive to gradient fidelity. A more comprehensive exploration of how compression affects different aspects of training—such as convergence speed, model accuracy, and robustness—would be informative for practitioners aiming to apply SAFL in resource-constrained environments.

**Questions:**

See Weakness

+ Error Feedback
Can you additionally compare w sketching + EF

+ Overall I feel like the paper is simply analysis of
Adaptive FL + Sketching + clipping ~
which was later packaged as a new algorithm.
Combining different techniques => Low Novelty.

---

### Meta-Review · Area_Chair_1MPW · 2024-12-22

**Metareview:**

This paper introduces the Sketched Adaptive Federated Learning (SAFL) framework, aiming to alleviate communication burdens in federated learning (FL) by leveraging gradient sketching alongside adaptive optimizers. The authors propose that by combining techniques like CountSketch and quantization with adaptive optimizers (e.g., Adam, AMSGrad), FL can achieve improved communication efficiency. The framework is extended to handle non-i.i.d. settings via a clipped variant, SACFL, which uses gradient clipping to ensure stability even with heavy-tailed noise. The paper’s convergence results are claimed to scale logarithmically with respect to model size, which is theoretically significant for applications with large model parameters. Additionally, experiments in vision and NLP domains provide empirical validation, showing that SAFL and SACFL can maintain comparable accuracy while achieving higher compression rates.


Main criticism:

1. Sketching belongs to the class of unbiased compressor with conically bounded variance, and hence all papers (hundreds of works) on communication compression using such compressors includes sketching as a special case. The authors do not compare to the key works in this very closely related literature. For example, for smooth nonconvex problems, the Marina method with the so-called permutation (PermK) compressor of Szlendak et al, Permutation Compressors for Provably Faster Distributed Nonconvex Optimization, arXiv:2110.03300, 202, achieves O(1/T) rate for the convergence of the square norm of the gradient, where the rate is independent of the dimension in the case when the number of workers ($C$) is larger than the number of dimensions ($d$)! In FL, the number of workers is very large. On the other hand, if $d>C$, the constant in the rate is $d/C$, and this will be equal to $\log d$ if $C = d/\log(d)$. This is true in the nonstochastic case - extensions to the stochastic case were considered in Gorbunov et al, MARINA: Faster Non-Convex Distributed Learning with Compression, ICML 2021 (see their VR-MARINA method which obtains $O(1/T)$ rate even in the stochastic case, unlike the $O(1/\sqrt{T})$ rate in the reviewed paper).

2. The authors claim, in several parts of the paper, including the abstract, that their results are competitive with the state of the art error feedback results. However, this claim is not substantiated in the paper, and, moreover, the authors do not even cite the SOTA EF methods, which are based on the EF21 method of Richtarik, Sokolov and Fatkhullin, EF21: A new, simpler, theoretically better, and practically faster error feedback, NeurIPS 2021. See, for example (there are many more follow-up works),
- Fatkhullin et al, EF21 with bells & whistles: practical algorithmic extensions of modern error feedback
- Richtarik et al, 3PC: Three point compressors for communication-efficient distributed training and a better theory for lazy aggregation, ICML 2022
- Fatkhullin et al, Momentum provably improves error feedback!, NeurIPS 2023
- Richtarik et al, Error feedback reloaded: from quadratic to arithmetic mean of smoothness constants, ICLR 2024

Granted, the above method are not adaptive. However, the claim that the communication complexity of all know methods has linear dependence on $d$ is not true; the question depends on the number of clients $C$ as well. In subsequent works, modern EF methods have been analyzed in the nonsmooth nonconvex case as well, with the assumption of bounded gradients, as in the reviewed paper.

3. The authors claim that error feedback is not needed when unbiased compressors are used. This is true, but in this case, one case use DIANA-type variance reduction [1, 2 & many follow-up works], which is just a different type of error-feedback applicable in this setting. The theoretical results of methods based on such VR mechanisms are vastly superior to the EF-based theory, which is necessarily worse since it handles a larger class of compressors.

- [1] Mishchenko et al, Distributed learning with compressed gradient differences, arXiv:1901.09269, 2019
- [2] Horvath et al, Stochastic distributed learning with gradient quantization and double variance reduction, arXiv:1904.05115, 2019

4. It is interesting that the results obtained by the authors depend in the Hessian spectrum. Since Newton's method can be seen as a prototypical "adaptive" method, and since communication-efficient variant of Newton method was proposed in the context of the FL literature before [3], with interesting theoretical (rates independent of condition number) and empirical results, comparison to this line of work seems to be called for:
- [3] Safaryan et al, FedNL: Making Newton-type methods applicable to federated learning, ICML 2022.

5. Other pieces of criticism were raised by the reviewers, but I am now running out of the character limit for this metareview.

The scores were mixed (3,3,6,6). In summary, I believe the paper is interesting, but needs a major revision.

**Additional Comments On Reviewer Discussion:**

Some good questions were asked by the reviewers (e.g., on scaling, assumptions, experiments, combination of error feedback and sketching), and I believe reasonable answers were given, in general.

---

Some more criticism raised by the reviewers (since I run out of character limit in the metareview):
- The theoretical results don't contain any new techniques, and the improved convergence rate seems to follow immediately from applying the eigenvalue decay assumption.
- It is not well-motivated to simultaneously assume bounded gradients (not satisfied by neural networks) and eigenvalue decay (motivated by neural networks). The authors address this in the rebuttal, but their response is somewhat incomplete.
- The experimental results from the original submission are weak, with many missing baselines. The authors added some new experiments in the rebuttal, but the presentation needs to be significantly improved.
- There is a disagreement with the claim by the authors of a connection between heterogeneous data and heavy-tailed noise in federated learning. This significantly weakens the motivation for the problem studied by this submission.

---

### Decision · Program_Chairs · 2025-01-22

Reject